# NOTCH signaling specifies arterial-type definitive hemogenic endothelium from human pluripotent stem cells

Gene I. Uenishi[1,2], Ho Sun Jung[1], Akhilesh Kumar [1], Mi Ae Park [1], Brandon K. Hadland[3,4], Ethan McLeod[1], Matthew Raymond[1,5], Oleg Moskvin [1], Catherine E. Zimmerman[1], Derek J. Theisen [1], Scott Swanson [6], Owen J. Tamplin [7], Leonard I. Zon [8], James A. Thomson[6,9,10], Irwin D. Bernstein[3,4] & Igor I. Slukvin [1,2,9]

NOTCH signaling is required for the arterial specification and formation of hematopoietic stem cells (HSCs) and lympho-myeloid progenitors in the embryonic aorta-gonad-mesonephros region and extraembryonic vasculature from a distinct lineage of vascular endothelial cells with hemogenic potential. However, the role of NOTCH signaling in hemogenic endothelium (HE) specification from human pluripotent stem cell (hPSC) has not been studied. Here, using a chemically defined hPSC differentiation system combined with the use of DLL1-Fc and DAPT to manipulate NOTCH, we discover that NOTCH activation in hPSC-derived immature HE progenitors leads to formation of CD144$^+$CD43$^-$CD73$^-$DLL4$^+$Runx1 + 23-GFP$^+$ arterial-type HE, which requires NOTCH signaling to undergo endothelial-to-hematopoietic transition and produce definitive lympho-myeloid and erythroid cells. These findings demonstrate that NOTCH-mediated arterialization of HE is an essential prerequisite for establishing definitive lympho-myeloid program and suggest that exploring molecular pathways that lead to arterial specification may aid in vitro approaches to enhance definitive hematopoiesis from hPSCs.

[1] Wisconsin National Primate Research Center, University of Wisconsin, Madison, WI 53715, USA. [2] Department of Pathology and Laboratory Medicine, University of Wisconsin School of Medicine and Public Health, Madison, WI 53792, USA. [3] Department of Pediatrics, University of Washington School of Medicine, Seattle, WA 98109, USA. [4] Clinical Research Division, Fred Hutchinson Cancer Research Center, Seattle, WA 98109, USA. [5] Department of Biomedical Sciences, University of Wisconsin School of Veterinary Medicine, Madison, WI 53706, USA. [6] Morgridge Institute for Research, Madison, WI 53715, USA. [7] Department of Pharmacology, University of Illinois, Chicago, IL 60612, USA. [8] Stem Cell Program and Division of Hematology/Oncology, Children's Hospital Boston, Harvard Medical School and Howard Hughes Medical Institute, Boston, MA 02115, USA. [9] Department of Cell and Regenerative Biology, University of Wisconsin School of Medicine and Public Health, Madison, WI 53707, USA. [10] Department of Molecular, Cellular, and Developmental Biology, University of California, Santa Barbara, CA 93106, USA. These authors contributed equally: Gene I. Uenishi, Ho Sun Jung. Correspondence and requests for materials should be addressed to I.I.S. (email: islukvin@wisc.edu)

Generating autologous hematopoietic stem cells (HSCs) from induced pluripotent stem cells (iPSCs) that can be precisely genetically modified with designer endonucleases, and subsequently clonally selected, represents a promising approach for patient-specific gene therapies. Although multiple studies were able to generate hematopoietic progenitors (HPs) with a HSC phenotype and limited engraftment potential from pluripotent stem cells (PSCs)[1–3], the robust and consistent engraftment with recapitulation of the full spectrum of terminally differentiated hematopoietic cells, including lymphoid cells, has not been achieved. Thus, identifying key cellular and molecular programs required for proper HSC specification in vitro is essential to overcome the current roadblocks.

During in vivo development, HSCs emerge by budding from hemogenic endothelium (HE) lining arterial vessels, primarily from the ventral wall of the dorsal aorta[4,5]. NOTCH signaling is essential for arterial specification and generation of HSCs[6,7]. Notch1$^{-/-}$, Dll4$^{-/-}$ and Rbpjk$^{-/-}$ mice, which are embryonic lethal, have severe impairment in arterial vasculogenesis, fail to develop the dorsal artery[6,8,9], and lack intra-embryonic hematopoiesis. NOTCH signaling is also required for the acquisition of arterial identity in extraembryonic vessels, including the yolk sac vasculature[10,11] and the specification of Flk-1$^+$c-Kit$^+$CD31$^+$CD45$^-$ hemogenic progenitors within yolk sac[12]. Notably, definitive HPs with lymphoid potential in the yolk sac, umbilical cord and vitelline vessels only emerge within the arterial vasculature[13,14]. In contrast, the primitive extraembryonic wave of erythropoiesis and the first wave of definitive yolk sac erythro-myelopoiesis (EMP), which lack lymphoid potential, are not NOTCH-depend or specific to the arterial vessels[6,9,13,15]. The lack of venous contribution to HSCs along with the shared requirements of Notch, VEGF, and Hedgehog signaling for both arterial fate acquisition and HSC development[16–19] led to the hypothesis that arterial specification could be a critical prerequisite for HSC formation. However, a direct progenitor-progeny link between arterial specification and definitive hematopoiesis has never been demonstrated. Moreover, demonstration in recent studies showing that HE represents a distinct CD73$^-$ lineage of endothelial cells[20,21] and that hematopoietic specification is initiated at the HE stage[22–24] raises the question whether NOTCH signaling at arterial sites creates a permissive environment for HSC development following endothelial-to-hematopoietic transition (EHT), or that arterial specification per se is required for HE to become HSCs. Although, recent studies have demonstrated that NOTCH activation induces arterialization of CD73$^+$ non-HE[21], and that NOTCH inhibition with DAPT reduces production of CD45$^+$ cells from CD34$^+$CD43$^-$CD73$^-$ HE progenitors[21,25], the effect of NOTCH signaling on HE specification has never been explored. Here, using a chemically defined human pluripotent stem cell (hPSC) differentiation system combined with the use of DLL1-Fc and the small molecule DAPT to manipulate NOTCH signaling following the emergence of the well-defined CD144$^+$CD43$^-$CD73$^-$ population of HE during EHT, we discover that NOTCH activation leads to the formation of arterial-type CD144$^+$CD43$^-$CD73$^-$DLL4$^+$ HE (AHE) that expresses arterial markers and possesses definitive lympho-myeloid and erythroid potentials. Using a transgenic reporter WA01, human embryonic stem cells (hESCs) in which the Runx1 + 23 enhancer mediates eGFP expression, we find that only DLL4$^+$, and not DLL4$^-$, HE cells, demonstrate enhancer activity that is typically found in HE at sites of definitive hematopoiesis in mouse and zebra fish embryos[26–28]. Hematopoiesis from CD144$^+$CD43$^-$CD73$^-$DLL4$^+$ AHE requires stroma and is strictly dependent on NOTCH activation. In contrast, NOTCH modulation has limited effect on EHT from the HE fraction that remains DLL4$^-$ following NOTCH activation, indicating that definitive hematopoietic activity segregates to AHE. Together, these studies establish a direct progenitor-progeny link between arterialization of HE and embryonic definitive hematopoiesis and reveal that NOTCH-mediated induction of AHE is an important prerequisite for establishing the definitive hematopoietic program from hPSCs.

## Results

**DLL1-Fc increases hematopoiesis and NOTCH signaling in HE.** In order to determine the direct effect of NOTCH signaling on hematoendothelial differentiation from hPSCs, we utilized a modified version of the serum- and feeder-free differentiation system described previously[29], where we identified developmental stage equivalencies to in vivo development that can be identified by cell-surface antigens and functional assays on specific days of differentiation; Day 2–3 APLNR$^+$PDGFRα$^+$ Primitive Mesoderm (D2 or D3 PM), Day 4 KDR$^{hi}$PDGFRα$^{low/-}$CD31$^-$ Hematovascular Mesoderm Precursors (D4 HVMP), Day 4 and 5 CD144$^+$CD43$^-$CD73$^-$ Hemogenic Endothelial Cells (D4 or D5 HE), and Day 8 CD34$^+$CD43$^+$ HPs (D8 HP)[20,29]. During differentiation, we found that the NOTCH1 receptor is first highly expressed on D4 HE cells while the NOTCH ligand, DLL4, is first expressed on D5 within the CD144$^+$ (VE-Cadherin) population (Fig. 1a), suggesting that NOTCH signaling in hPSC cultures is established at the time of HE formation.

Following the establishment of optimal conditions for EHT culture in defined feeder- and serum-free conditions, we isolated D4 HE by magnetic enrichment of CD31$^+$ cells, since at this stage (Fig. 1a), the CD31$^+$ population is entirely CD144$^+$CD43$^-$CD73$^-$DLL4$^-$ [20,29] (Supplementary Fig. 1a). Isolated D4 HE cells were cultured either in control conditions, with the small molecule gamma-secretase inhibitor, DAPT, to inhibit NOTCH signaling, or were plated onto the immobilized NOTCH ligand DLL1-Fc to activate NOTCH signaling[30,31] (Fig. 1b). As we confirmed by western blot analysis of the active form of NOTCH1, NOTCH: ICD, and qPCR analysis of the downstream NOTCH1 target gene, HES1, by qPCR, these respective conditions efficiently inhibited and activated NOTCH signaling (Fig. 1c, d). Kinetic analysis of CD144 (endothelial marker) and CD43 (hematopoietic marker) from D4 + 1 to D4 + 4 reveals a significant increase in hematopoiesis in the NOTCH activation condition and a significant decrease in hematopoiesis in the NOTCH inhibition condition, compared to control (Fig. 1e). These results were consistent with other hESC and hiPSC lines (Supplementary Fig. 1b). In addition, similar results were obtained when D4 HE cells are cultured in serum-containing medium on wild-type OP9 stromal cells or OP9 cells transduced with human DLL4 (OP9-DLL4; Supplementary Fig. 1c). We also observe a significant increase in the total hematopoietic cell number in the NOTCH activation condition (Fig. 1f, two-way ANOVA, Bonferroni post-hoc test). The effect of DLL1-Fc on hematopoiesis increased as the concentration of immobilized DLL1-Fc and cell density increased (Supplementary Fig. 1e). In contrast, culture of D4 HE on immobilized JAG1-Fc or OP9-JAG1 minimally affected hematopoiesis as compared to controls (Supplementary Fig. 1d, f), thereby suggesting suboptimal activation of NOTCH signaling by JAG1.

**NOTCH activation facilitates EHT in HE.** The increase in hematopoiesis due to increased NOTCH signaling can be attributed to three reasons: (1) increased EHT, (2) increased hematopoietic expansion, or (3) increased survival post-EHT. To evaluate these possibilities, we isolated D4 HE cells and cultured them with DAPT for either 1 day during initiation of EHT (from D4 to D4 + 1), or throughout the entire culture (D4 to D4 + 4), followed by kinetic analysis of CD43 and CD144 expression on

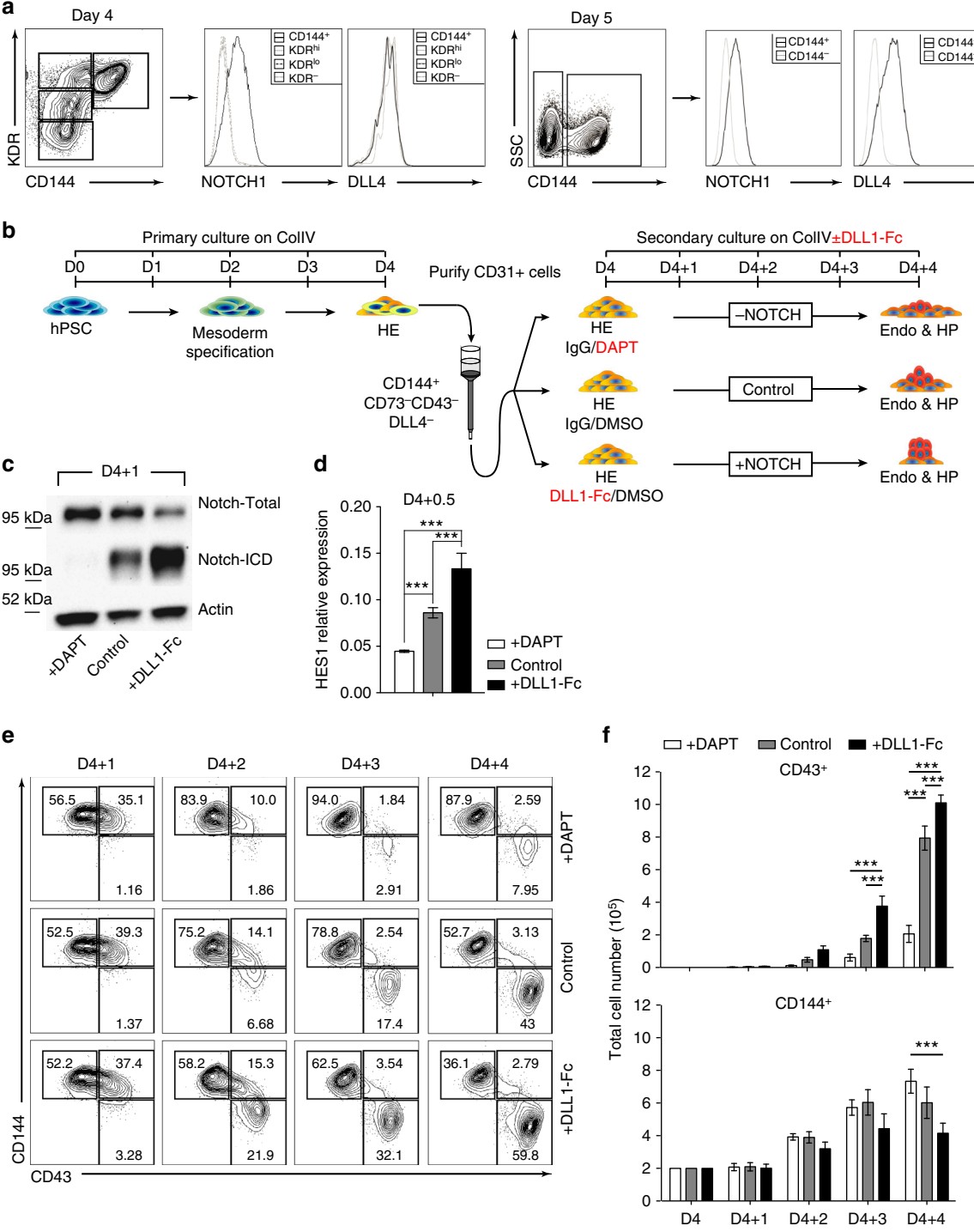

**Fig. 1** NOTCH activation increases hematopoiesis from D4 HE. **a** NOTCH1 receptor expression is first detected on D4 CD144+ cells. DLL4 expression is first detected on D5 CD144+ endothelial cells. **b** Schematic diagram of experiments. Cells were differentiated for 4 days on collagen IV, D4 CD144+CD43−CD73− HE cells were purified using CD31 microbeads and plated in 3 different NOTCH conditions. **c** Western blot of D4 HE cultured for 24 h (D4 + 1) in presence of DAPT or DLL1-Fc shows a decrease in the activated cleaved form of NOTCH1 in DAPT-treated cells, and an increase in the activated cleaved form of NOTCH1 in cells plated on DLL1-Fc. **d** qPCR analysis shows decreased *HES1* mRNA expression in D4 HE cultured for 12 h (D4 + 0.5) with DAPT, while *HES1* mRNA expression is increased in cells plated on DLL1-Fc. Results are mean ± s.e.m. for three independent experiments; two-way ANOVA Bonferroni post-hoc test, ***$p < 0.01$. **e** Flow cytometry on each day from D4 + 1 to D4 + 4 shows decreased CD43+ HPs in the cultures treated with DAPT, and increased HPs in the cultures plated on DLL1-Fc. Representative contour plots from three independent experiments are shown. **f** Total numbers of CD43+ HPs and CD144+CD43− endothelial cells in cultures plated on DLL1-Fc. Results are mean ± s.e.m. for three independent experiments; two-way ANOVA Bonferroni post-hoc test, ***$p < 0.001$

each day of the culture period (Fig. 2a). Following culture in defined conditions, HE weakly upregulate CD43 expression on D4 + 1, but retain flat endothelial morphology. Round CD43hi cells that have completed EHT appear after D4 + 2[32]. As shown in Fig. 2b, c, HE treated for 24-hours with DAPT from D4 to D4 + 1 weakly express CD43 along with CD144 on D4 + 1, but fail to complete EHT efficiently, as evidenced by a significant drop in CD43hiCD144− cells on D4 + 2 through D4 + 4, although DAPT treatment throughout (D4 to D4 + 4) more profoundly decreased hematopoiesis (two-way ANOVA, Bonferroni post-hoc test).

To further verify that NOTCH activation affects EHT, we also performed a single-cell deposition assay of the D4 HE using the OP9 stromal cells and serum-containing medium that support hematoendothelial development from single cells. Using a DOX-inducible DLL4 OP9 cell line (OP9-iDLL4), we deposited D4 HE onto 96-well plates of three different conditions; OP9-iDLL4 with DAPT without DOX-pretreatment (NOTCH inhibition condition), OP9-iDLL4 with DMSO without DOX-pretreatment (control condition), and OP9-iDLL4 with DMSO with pretreatment of DOX (NOTCH activation condition). We found that D4

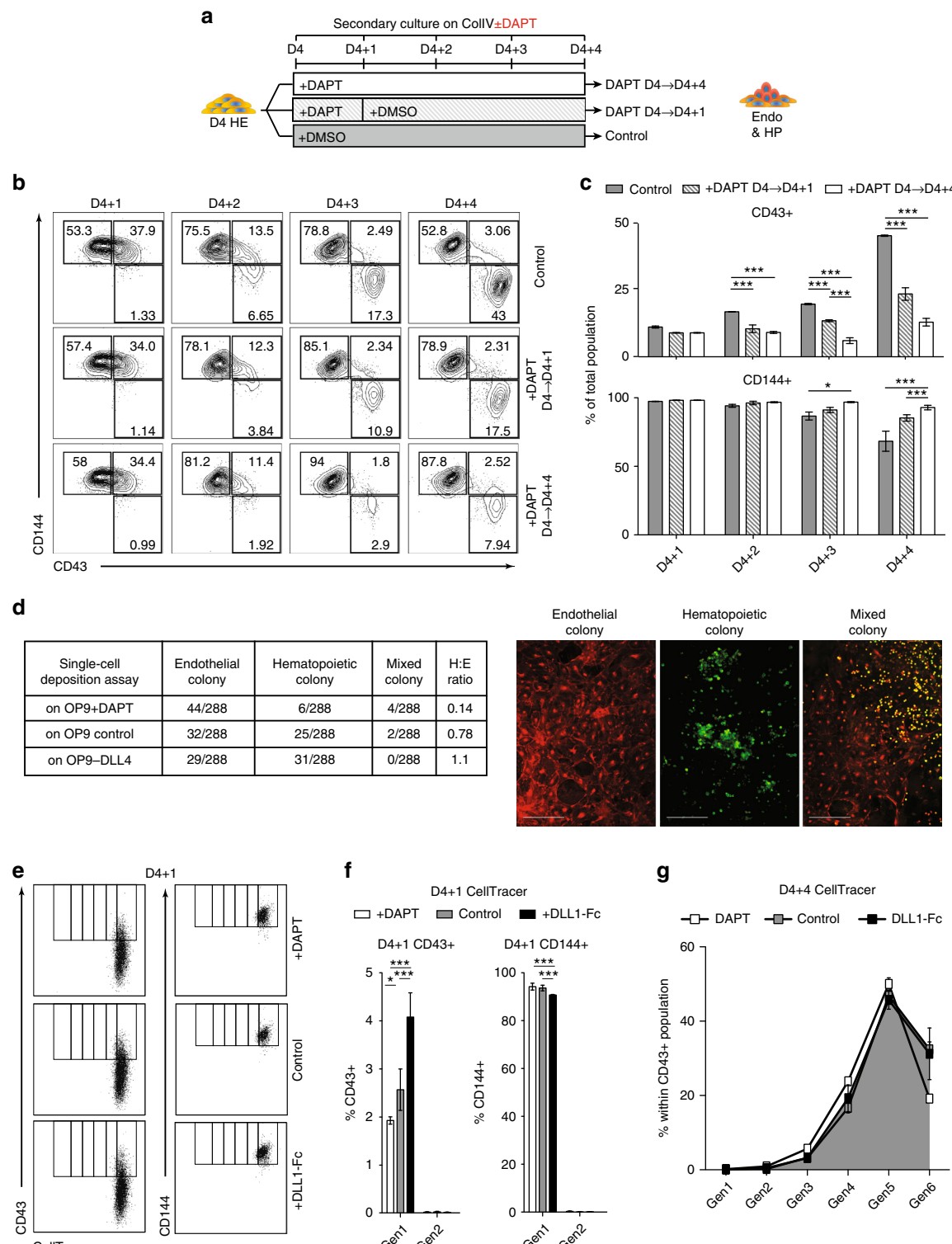

HE in the NOTCH inhibition condition had a markedly decreased ratio of hematopoietic/endothelial colonies compared to D4 HE cells in the control condition. In contrast, the D4 HE in the NOTCH activation condition had substantially increased ratio of hematopoietic colonies compared to D4 HE in the NOTCH inhibition condition, and a slight increase compared to D4 HE in the control condition (Fig. 2d). Due to the well-recognized fragility of hPSC-derived HE and survival after single-cell sorting[1,21], we found that less than 20% of single cells form endothelial or hematopoietic colonies. Nevertheless, the total number of colonies was consistent across each of the three NOTCH conditions, indicating that the sorting experiments were not affected by differences in cell viability.

We also stained the purified D4 HE before plating with CellTracer to track cell proliferation. When we analyzed the cells in each of the three NOTCH conditions on D4 + 1, we found that there was a statistically significant increase in the proportion of $CD144^+CD43^+$ to $CD144^+CD43^-$ cells within the first generation of cells in the NOTCH activation condition (+DLL-Fc), when compared to the NOTCH inhibition (+DAPT) condition (two-way ANOVA, Bonferroni post-hoc test). This result, in combination with the absence of a second generation on D4 + 1, suggests that the activation of NOTCH signaling at HE stage potentiate EHT initiation, but not proliferation (Fig. 2e, f). Assessment of cell proliferation on D4 + 4 with CellTracer in cultures treated with DAPT through D4 + 4 revealed no significant shift in distribution of $CD43^+$ cells within each generation (Fig. 2g and Supplementary Fig. 2a), consistent with the lack of NOTCH effect of post-EHT expansion. In addition, analysis of cell cycle in these cultures using EdU, demonstrated no differences in cycling $CD43^+$ cells in the different NOTCH conditions (Supplementary Fig. 2b, c).

To evaluate whether NOTCH signaling affects apoptosis, we performed Annexin V flow cytometric analysis of HE cultured with DAPT, DMSO, or on DLL1-Fc on D4 + 4. As shown in Supplementary Fig. 3a, b, none of the conditions affected apoptosis of blood cells post-transition, suggesting that the difference in hematopoiesis from HE following manipulation of NOTCH signaling is not attributed to the NOTCH effect on cell survival.

Together, these results suggest that NOTCH activation at the HE stage facilitates EHT, but has minimal effect on expansion or survival of blood cells at the post-EHT stage.

**NOTCH activation increases definitive hematopoiesis from HE.** Next, we determine whether NOTCH has an effect on HPs emerging through EHT. Although NOTCH1 expression decreases among the $CD144^+$ endothelial population from D4 + 1 to D4 + 4, $CD144^-CD43^+$ blood cells increase and maintain expression of NOTCH1 post-transition from D4 + 2 to D4 + 4, notably among the $CD34^+$ subpopulation (Supplementary Fig. 4a,b), thereby indicating that emerging blood cells are equipped to respond to NOTCH signaling. To determine how NOTCH affects post-EHT hematopoietic differentiation, we collected cells from D4 + 4 HE cultures from the 3 different NOTCH conditions (DAPT, DMSO, or DLL1-Fc) and plated them in methylcellulose to measure their colony forming potential. The total number of colonies was significantly lower in the DAPT-treated NOTCH inhibition condition, while there was no significant change in the total number of colonies between the control condition and the NOTCH activation condition. Critically, however, there was a statistically significant increase in multipotent GEMM-CFCs and GM-CFCs, as well as in E-CFCs among the HP cells from the HE cultured in NOTCH activation condition compared to control (Fig. 3a, two-way ANOVA, Bonferroni post-hoc test). These results suggest that NOTCH activation maintains multilineage potential of emerging HPs.

Next, we determined whether increased NOTCH activation increases definitive-type hematopoiesis. Previously, the Runx1 + 23 enhancer was found to be active in all HPs, including yolk sac. HE found in regions where definitive hematopoiesis emerges have also been found to activate Runx1 + 23, including the para-aortic splanchnopleura, AGM region, vitelline, and umbilical arteries[26–28,33,34]. We generated a hESC reporter line with Runx1 + 23 enhancer driving eGFP expression knocked into the AAVS1 locus (Supplementary Fig. 5a, b). We differentiated the Runx1 + 23 cell line, purified the D4 HE cells, and plated them in each of the 3 NOTCH conditions. Critically, there was significantly higher eGFP expression from D4 + 1 to D4 + 4 that emerge from the $CD144^+$ population in the NOTCH activation condition compared to the control. In contrast, cells treated with DAPT (NOTCH inhibition) had less eGFP expression compared to the control (Fig. 3b).

T-cell potential is another hallmark of definitive hematopoiesis[35]. Comparative analysis of T-cell potential of the D4 + 4 $CD43^+$ cells from DAPT, DLL1-Fc, and control conditions revealed that HPs from the NOTCH inhibition condition had no T-cell potential, while HPs from the NOTCH activation condition had significantly increased T-cell potential compared to HPs from the control condition (Fig. 3c, one-way ANOVA Bonferroni post-hoc test). In a separate assay, we collected floating HPs on D4 + 4 and continued culture in a modified erythrocyte expansion condition[36]. After 10 days, we collected the cells and isolated mRNA to analyze their globin expression. We found that erythrocytes generated from HPs from the NOTCH activation condition have significantly increased ratios of adult-type β-globin expression to embryonic ε-globin and fetal γ-globin expression, and the ratio of adult-type α-globin expression to embryonic ζ-globin expression, when compared to the erythrocytes generated from HPs from both the NOTCH inhibition condition and the control condition (Fig. 3d, one-way ANOVA

**Fig. 2** Increased NOTCH activation facilitates EHT. **a** Schematic diagram of experiments. D4 HE cultured in presence of DAPT for 4 days (D4 + 4) or 1 day (D4 + 1), or DMSO (control). $CD144^+$ endothelial and $CD43^+$ blood cells were analyzed following 4 days of culture. **b** Flow cytometric analysis demonstrates that NOTCH activation facilitates EHT as evidenced by the decrease in hematopoietic activity when DAPT is added only from D4 to D4 + 1. Representative contour plots from three independent experiments are shown. **c** Frequencies of endothelial and blood cells in HE cultures treated with DAPT or DMSO (control). Results are mean ± s.e.m. for at least three independent experiments; two-way ANOVA Bonferroni post-hoc test, *$p < 0.05$, ***$p < 0.001$. **d** Single D4 HE cells were FACS-sorted into 96-well plate with OP9, OP9 + DAPT, and OP9-DLL4. Colonies were scored based on CD43 and CD144 expression on D4 + 10 by immunofluorescence and counted by eye. Scale bar represents 100 μm. **e** Representative flow cytometric cell proliferation analysis and **f** bar graph conducted with CellTracer shows an increase in the first generation (Gen1) $CD43^+$ cells on D4 + 1 and a proportional decrease in Gen1 $CD144^+$ endothelial cells, suggesting that the increase in blood cells is due to an increase in EHT and not just proliferating HPs. **g** Line graph depicting the percent of each generation within the $CD43^+$ population on D4 + 4 in each of the NOTCH treatment conditions. Results are mean ± s.e.m. for three independent experiments; two-way ANOVA Bonferroni post-hoc test, *$p < 0.05$, ***$p < 0.001$. No significant change of each generation between conditions suggesting that NOTCH does not affect proliferation of HPs. Generation gates in **f**, **g** were determined by concatenating D4 to D4 + 4 results and utilizing FlowJo's proliferation assay

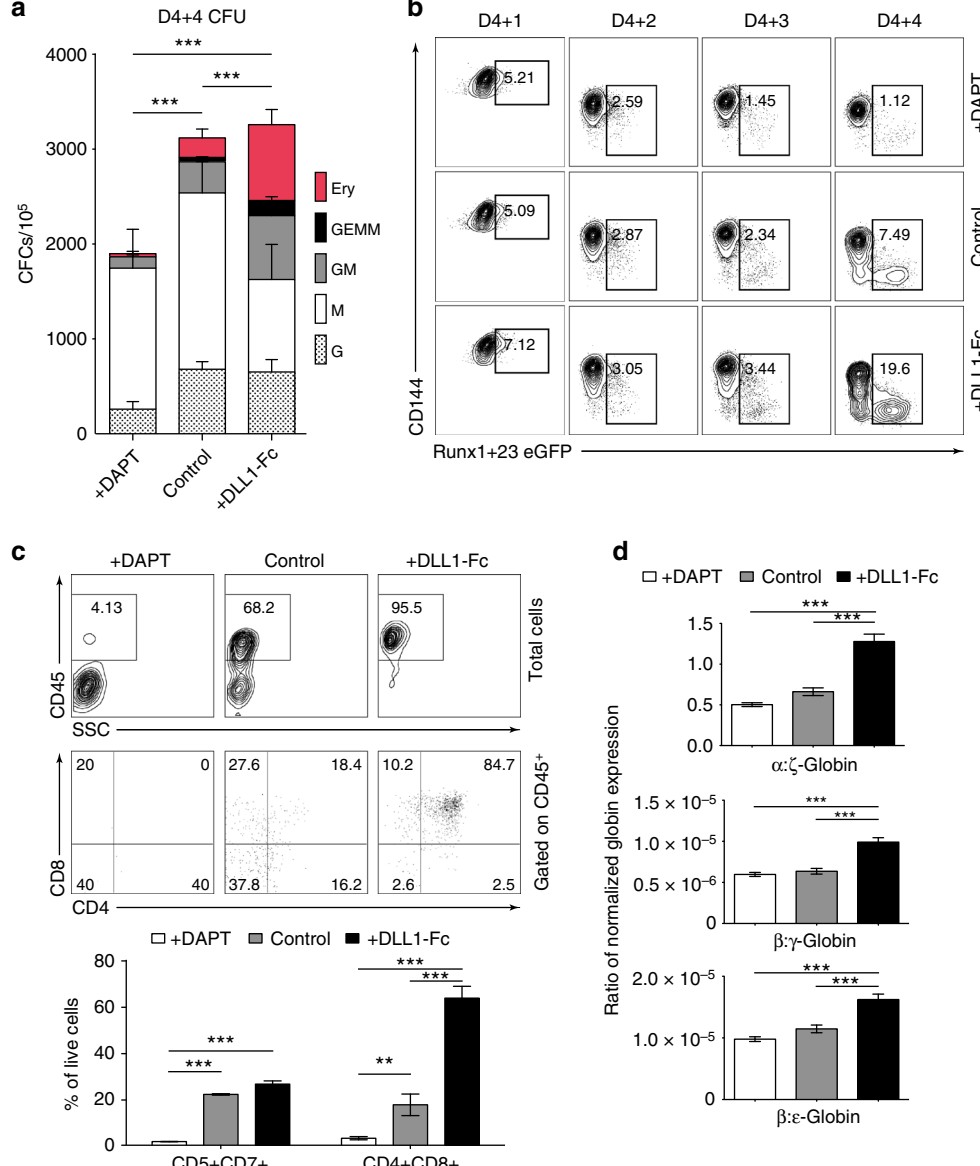

**Fig. 3** NOTCH activation at HE stage increases definitive hematopoiesis. **a** D4 HE were cultured with DAPT or in the presence of DLL1-Fc (see Fig. 1b schematic diagram). Cells were collected after 4 days of differentiation (D4 + 4) and used to determine frequencies of hematopoietic progenitors in CFU assay. Increase in multipotential GEMM and GM colonies in the DLL1-Fc culture condition suggests that NOTCH activation supports expansion of the most immature HPs. Results are mean ± s.e.m. for three independent experiments; two-way ANOVA Bonferroni post-hoc test, ***$p < 0.001$ **b** Flow cytometric analysis of Runx1 + 23-eGFP transgene expression in D4 HEPs cultured with DAPT or on DLL1-Fc. Runx1 + 23 enhancer activity increases in the cultures plated on DLL1-Fc and decreases in the DAPT-treated cultures. Representative contour plots from three independent experiments are shown. **c** T-cell potential of HP collected after 4 days of culture D4 HEs in presence of DAPT or DLL1-Fc. Bars show mean ± s.e.m. for three independent experiments; one-way ANOVA Bonferroni post-hoc test, **$p < 0.01$, ***$p < 0.001$. **d** Ratio of α/ζ, β/γ, and β/ε globin chain expression in erythroid cultures generated from D4 HE in presence of DAPT or DLL1-Fc. Results are mean ± s.e.m. for three independent experiments; one-way ANOVA Bonferroni post-hoc test, ***$p < 0.001$

Bonferroni post-hoc test). Overall, these findings suggest that NOTCH signaling is required for definitive hematopoietic stem/progenitor cell specification.

**NOTCH activation induces DLL4+ HE with arterial identity**. Previously, we identified CD73 expression to demark the loss of hemogenic potential within the D5 CD144+ endothelial population[20]. As we demonstrated above, D4 HE cells lacked the expression of the arterial marker, DLL4. However, when we analyze CD73 and DLL4 expression within the D4 + 1 and D4 + 2 CD144+ populations in each of the three NOTCH conditions,

we found a significant increase in a unique transient population of CD73−DLL4+ endothelial cells in the NOTCH activation condition, and a delayed upregulation of CD73 expression on DLL4+ endothelial cells compared to the NOTCH inhibition and control conditions (Fig. 4a, b, two-way ANOVA Bonferroni post-hoc test). In addition, when we analyze the CD144+ population of the Runx1 + 23 cell line on D4 + 1, we found that all eGFP+ cells are within the CD144+CD73−DLL4+ population (Fig. 4c and Supplementary Fig. 5c). Since DLL4 is expressed by HE underlying intra-aortic hematopoietic clusters in the AGM[37], our results suggest that the DLL4+ population may resemble arterial-type definitive HE found in arterial vasculature. To corroborate

this hypothesis, we evaluated the expression of arterial, venous, and definitive hematopoietic markers by real-time qPCR analysis of sorted D4 CD144$^+$CD43$^-$CD73$^-$HE that are DLL4$^-$ by default (D4 HE) and D5 CD144$^+$ endothelial subpopulations CD144$^+$CD43$^-$CD73$^-$DLL4$^+$ (D5 HE:DLL4$^+$), CD144$^+$CD43$^-$CD73$^-$DLL4$^-$ (D5 HE:DLL4$^-$), and CD144$^+$CD43$^-$CD73$^+$DLL4$^-$ (D5 nonHE:DLL4$^-$), (Fig. 4d). This analysis reveals that the D5 HE:DLL4$^+$ and nonHE:DLL4$^+$ populations have increased expression of *NOTCH1, DLL4, EFNB2, HEY2, SOX17,* and *CXCR4* genes associated with arterial endothelium, and decreased expression of *NR2F2* associated with venous endothelium, when compared to D5 DLL4$^-$HE and nonHE populations.

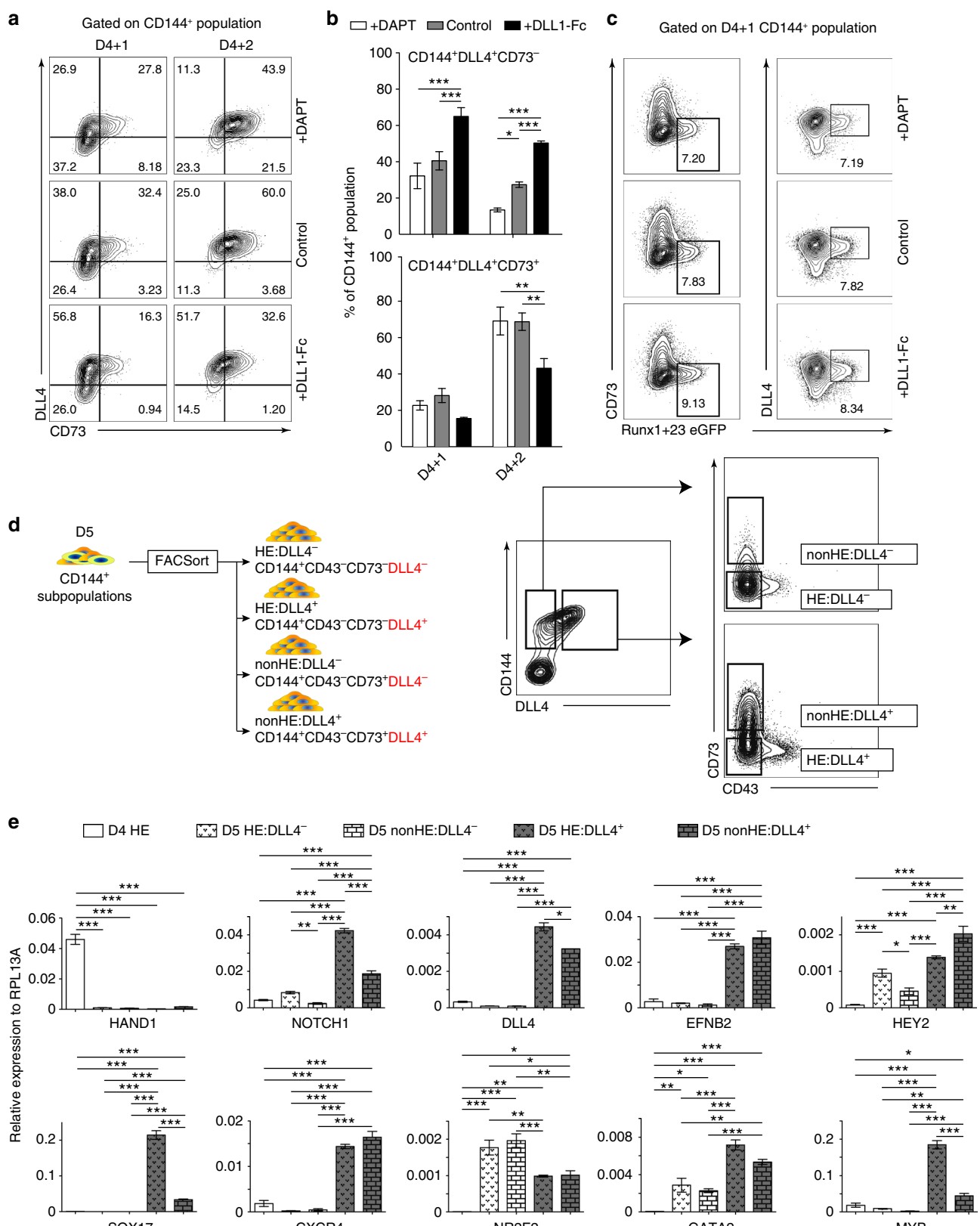

In contrast, D5 HE:DLL4⁻ demonstrated an increased expression of *NR2F2* venous marker. Importantly, genes associated with definitive hematopoiesis, *MYB* and *GATA2*, were expressed significantly higher in the D5 HE:DLL4⁺ population compared to the D5 HE:DLL4⁻ population and D5 nonHE:DLL4⁺ populations (Fig. 4e, one-way ANOVA Bonferroni post-hoc test). We also revealed that emerging D4 HE cells that are lacking DLL4 expression were different from D5 HE:DLL4⁻ and D5 HE:DLL4⁺ cells. D4 HE did not express significant levels of arterial and venous markers, but retained expression of *HAND1*, which is expressed in extraembryonic and lateral plate mesoderm[38], suggesting that D4 HE may represent immature HE cells.

**Definitive HPs emerge from arterial-type HE.** To determine the hematopoietic potential of endothelium with arterial identity, we continued differentiation of hPSCs to D5, and then sorted the D5 CD144⁺CD43⁻CD73⁻DLL4⁻ (HE:DLL4⁻) and D5 CD144⁺CD43⁻CD73⁻DLL4⁺ (HE:DLL4⁺) endothelial subpopulations (Fig. 5a). Although we did not detect any CD43⁺ blood cells from D5 HE:DLL4⁺ on D5 + 4 in serum- and feeder-free conditions with or without DLL1-Fc, these cells did produce blood when plated on OP9-DLL4 (Fig. 5b). In contrast, D5 HE:DLL4⁻ cells undergo EHT and develop HPs on D5 + 4 on both OP9 and OP9-DLL4. However, unlike previously, when we used D4 HE cells before they bifurcate into D5 HE:DLL4⁺ and D5 HE:DLL4⁻, there was no significant difference in blood production between the D5 HE:DLL4⁻ cells plated on OP9 vs. OP9-DLL4 (Fig. 5b, c, two-way ANOVA Bonferroni post-hoc test). The results were consistent across different iPSC lines as well (Supplementary Fig. 6). In addition, DAPT treatment from D5 to D5 + 2, D5 + 2 to D5 + 4, and from D5 to D5 + 4 significantly inhibited hematopoietic activity of the HE:DLL4⁺ population, while DAPT treatment of HE:DLL4⁻ cultures had no effect on hematopoietic activity (Fig. 5d, two-way ANOVA Bonferroni post-hoc test), suggesting that hematopoiesis from D5 HE:DLL4⁺, in contrast to D5 HE:DLL4⁻, is NOTCH dependent.

Next, we determined whether the HPs from each of the D5 HE subsets have differential definitive hematopoietic potential. When the HPs from the D5 HE subpopulations were plated in colony forming medium, the HPs that emerged from the HE:DLL4⁺ subpopulation cultured on OP9-DLL4 had increased colony forming cells, particularly of GEMM-CFCs compared to the HPs from D5 HE:DLL4⁻ on OP9 and OP9-DLL4 (Fig. 5e, two-way ANOVA Bonferroni post-hoc test).

When we collected the floating HPs derived from D5 HE:DLL4⁻ on OP9 and OP9-DLL4, and HPs derived from D5 HE:DLL4⁺ on OP9-DLL4, and continued to grow them in the aforementioned erythrocyte expansion and maturation culture[36], we found that erythrocytes generated from HPs derived from the D5 HE:DLL4⁺ on OP9-DLL4 have significantly increased ratios of β-globin expression to ε-globin and γ-globin expression, and an increased ratio of α-globin expression to ζ-globin expression, when compared to the erythrocytes generated from HPs derived from D5 HE:DLL4⁻ on OP9 and OP9-DLL4 (Fig. 5f, one-way ANOVA Bonferroni post-hoc test).

We also conducted a limiting dilution assay (LDA) for lymphoid potential, and found that 1 in 14 HPs derived from D5 HE:DLL4⁺ on OP9-DLL4 have T-cell potential, while 1 in 44 HPs derived from D5 HE:DLL4⁻ on OP9-DLL4 have lymphoid potential. Crucially, HPs derived from D5 HE:DLL4⁻ on OP9 and D5 HE:DLL4⁻ on OP9 with DAPT had only 1 in 10,706 and 1 in 10,895 had T-cell potential, respectively (Fig. 5g), thereby suggesting that D5 HE:DLL4⁺ phenotype enriches for HE that can produce HPs with T lymphoid potential.

In order to determine whether there are any molecular differences between HPs derived from HE:DLL4⁺ and HE:DLL4⁻ cells, we performed RNA-seq analysis of CD235a/CD41a⁻CD34⁺CD43⁺CD45⁺ cells generated from these two different hemogenic endothelial cells following tertiary culture on either OP9 or OP9-DLL4 (Fig. 6a). As a basis for the analysis, genes that were differentially expressed in a three-way Bayesian model involving HPs from HE:DLL4⁻ on OP9 (condition 1), HE:DLL4⁻ on OP9-DLL4 (condition 2), and HE:DLL4⁺ on OP9-DLL4 (condition 3) were used with focus specifically on genes upregulated in HE:DLL4⁺ vs. HE:DLL4⁻-derived HPs obtained from OP9-DLL4 cocultures. Among 131 differentially expressed genes in this category (Supplementary Dataset 1), we identified two cell-surface markers of HSCs in AGM, *ACE*, and *TEK*, and the following nine transcription factors: *MECOM*, *GFI1B*, and *ERG*, essential for AGM and fetal liver hematopoiesis; *ARID5B*, *BCOR*, and *KDM6B*, control lymphoid development[39–41]; *ZNF93*, highly expressed in T-cells[42]; and *RUNX1T1* and *HOXB8*, regulate expansion of blood progenitors[43,44] (Fig. 6b). Using the known transcription-target relationships obtained by combining largely complementary data from HTRIdb[45] and CellNet[46], 163 regulatory interactions involving 110 transcription factors upstream of the nine differentially expressed transcription factor-encoding genes were pulled to construct a regulatory network in HPs derived from HE:DLL4⁺ cells on OP9-DLL4 (Fig. 6c). The database-derived structure of the network has been confirmed by our RNA-Seq data: transcription factors that are active according to our regulon analysis (red nodes) are apparently responsible for the upregulation of mRNA level of the target genes (large nodes). Three out of nine target genes (*MECOM*, *RUNX1T1*, and *GFI1B*) have also an evidence of their protein-level activity (reddish color on the graph) detected as enrichment of their known targets among the differentially expressed genes. Importantly, *GATA2*, *SOX17*, *SOX18*, *MYB*, *PBX1*, *PRDM14*, *DACH1*, *KLF4*, *HOXA5*, *HOXA7*, and *NOTCH1* were identified as upstream regulators of these genes, thereby suggesting that the molecular program in HPs derived from the arterial-type HE:DLL4⁺ is driven by transcriptional regulators implicated in arterial development and definitive hematopoiesis.

To further verify the link between arterialization and definitive hematopoiesis, we assessed whether lymphoid development was affected by arterial gene *SOX17*, which is indispensable for acquisition and maintenance of arterial identity in the embryo[47]. For this purpose, we transfected hESC differentiation cultures with SOX17 siRNA at initiation of HE formation to achieve up to 50% reduction of SOX17 mRNA expression 48 h after treatment

**Fig. 4** NOTCH activation induces formation of arterial-type HE cells. **a** Flow cytometric analysis of DLL4 and CD73 expression following culture of D4 HE for 1 or 2 days in the presence of DAPT or DLL1-Fc. NOTCH activation on D4 HE specifically increases the CD144⁺CD73⁻DLL4⁺ population. **b** Frequencies of DLL4⁺ cells in hemogenic (CD73⁻) and non-hemogenic fractions of endothelium following 1 and 2 days of culture of D4 HE in the presence of DAPT or DLL1-Fc. Results are mean ± s.e.m. for three independent experiments; two-way ANOVA Bonferroni post-hoc test, *p < 0.05, **p < 0.01, ***p < 0.001. **c** Flow cytometric analysis of Runx1 + 23 enhancer activity following 1 day of culture of D4 HE in presence of DAPT or DLL1-Fc. Runx1 + 23 enhancer activity is limited to the CD144⁺CD73⁻DLL4⁺ population. Representative contour plots from two independent experiments are shown. **d** Schematic diagram of FACS isolation of endothelial subpopulations formed on D5 of differentiation. **e** qPCR analysis of arterial (*NOTCH1*, *DLL4*, *EFNB2*, *HEY2*, *SOX17*, *CXCR4*), venous (*NR2F2*), hematopoietic (*MYB*, *GATA2*), and mesodermal (*HAND1*) genes in D4 HE and D5 endothelial subpopulations. Results are mean ± s.e.m. for three independent experiments; one-way ANOVA Bonferroni post-hoc test, *p < 0.05, **p < 0.01, ***p < 0.001

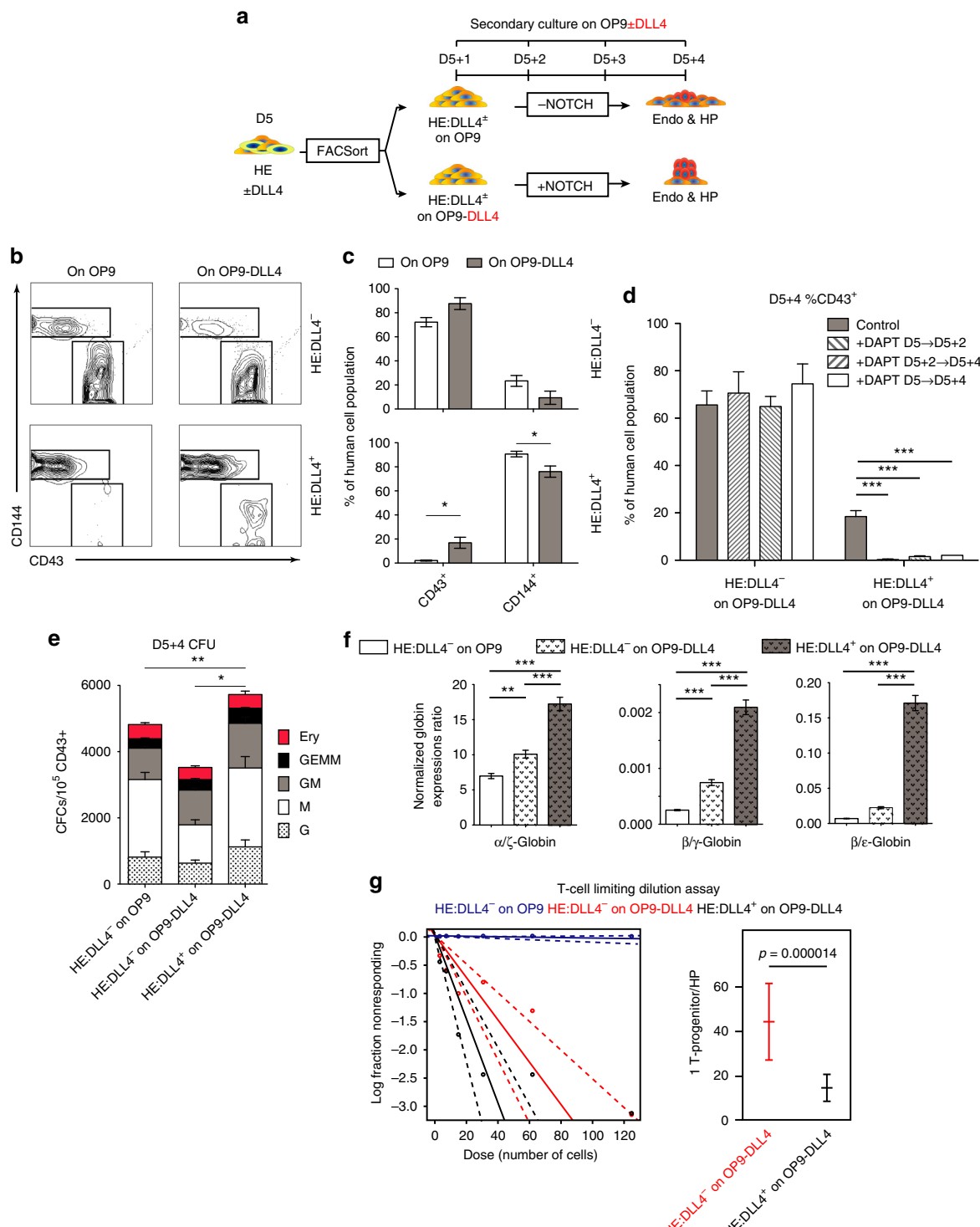

**Fig. 5** Arterial-type HE undergoes EHT under high NOTCH activation and produce definitive-type HPs. **a** Schematic diagram of subsequent experiments. D5 CD144$^+$CD43$^-$CD73$^-$ were sorted based on DLL4 expression (D5 HE:DLL4$^{+/-}$) using FACS and cultured on either OP9 or OP9-DLL4 for 4 days (D5 + 4). **b**, **c** Flow cytometric analysis of CD43$^+$ hematopoietic and CD144$^+$ endothelial cells following culture of D5 HE:DLL4$^+$ and D5 HE:DLL4$^-$ on either OP9 or OP9-DLL4. Bars in **c** are mean ± s.e.m. for at least three independent experiments. Two-way ANOVA Bonferroni post-hoc test, *$p < 0.05$. **d** The effect of NOTCH inhibition with DAPT on blood production from D5 DLL4$^+$ and DLL4$^-$ HE. No significant differences were found when HE:DLL4$^-$ cells were treated with DAPT. Results are mean ± s.e.m. for three independent experiments. Two-way ANOVA Bonferroni post-hoc test, ***$p < 0.001$. **e** CFC potential of hematopoietic cells generated from D5 DLL4$^+$ and DLL4$^-$ HE following 5 days culture on OP9-DLL4. Results are mean ± s.e.m. for at least three independent experiments; two-way ANOVA Bonferroni post-hoc test, *$p < 0.05$, **$p < 0.01$. CFC-GEMMs are significantly increased in DLL4$^+$ cultures on OP9-DLL4. **f** Ratio of α/ζ, β/γ, and β/ε globin chain expression in erythroid cultures generated form hematopoietic cells collected from D5 DLL4$^+$ and DLL4$^-$ HE cultured on OP9-DLL4 (D5 + 4 cells). Results are mean ± s.e.m. for three independent experiments. one-way ANOVA Bonferroni post-hoc test, **$p < 0.01$, ***$p < 0.001$. **g** Limiting dilution assay to determine the frequency of T-cell progenitors within the D5 + 5 HPs generated from HE: DLL4$^-$ on OP9, HE:DLL4$^-$ on OP9-DLL4, and HE:DLL4$^+$ on OP9-DLL4

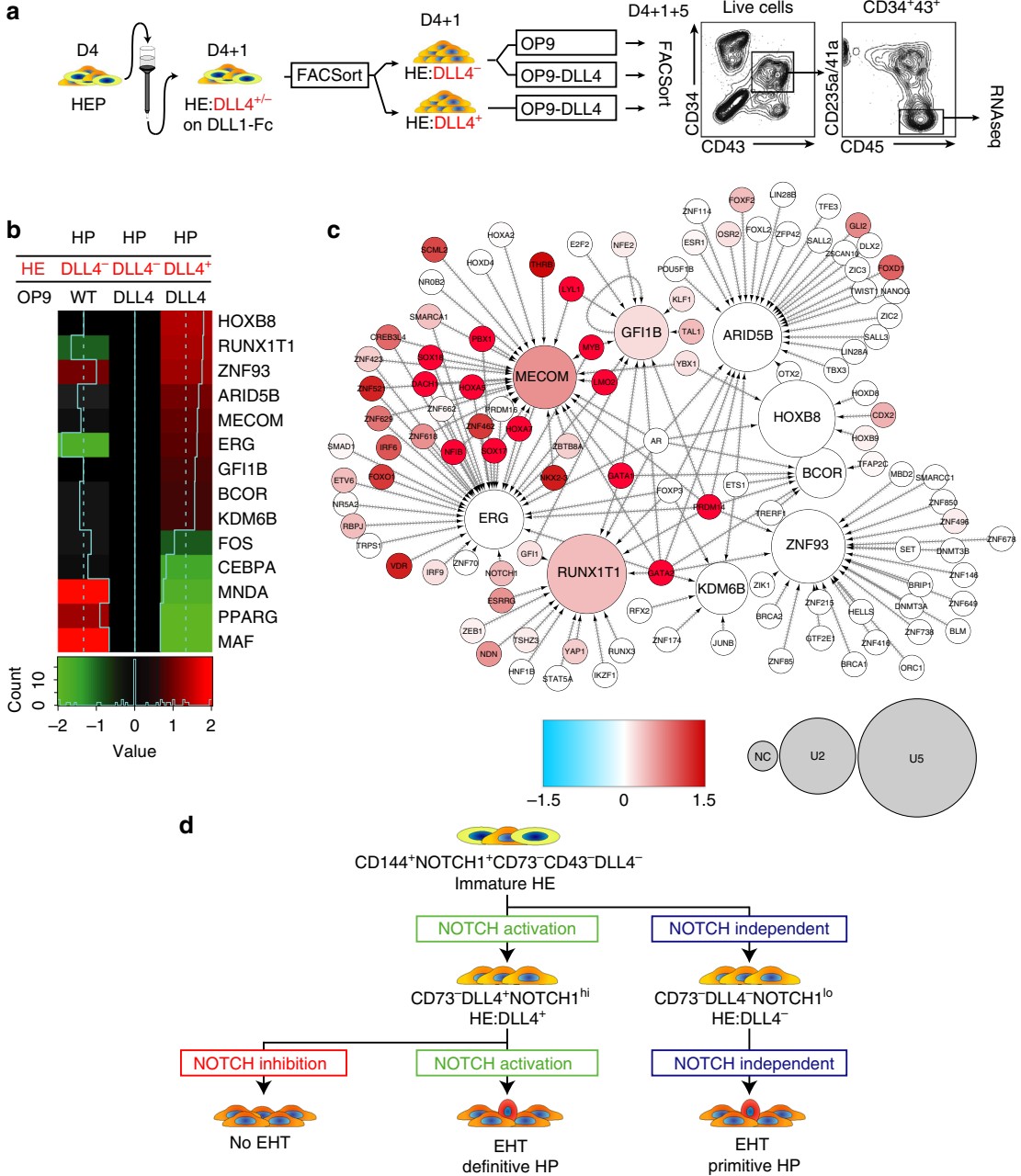

**Fig. 6** HPs derived from DLL4$^+$ HE cells activate definitive hematopoietic program. **a** Experimental strategy for generating and characterizing HE:DLL4$^{+/−}$-derived HPs. D4 HE cells where cultured on DLL1-Fc for 24 h, followed by purification of D4 + 1 HE:DLL4$^+$ and HE:DLL4$^−$ and subsequent culture on OP9 or OP9-DLL4. Five days later (D4 + 1 + 5), CD34$^+$CD43$^+$CD45$^+$CD235a/41a$^−$ population was FACSorted from each condition and RNA was extracted for RNA-seq. **b** A heatmap of differentially expressed transcription factor genes in HPs derived from indicated cell populations. The expression is shown as a log ratio of gene expression relative to HPs generated from HE:DLL4$^−$ cells on OP9-DLL4. **c** Transcriptional regulatory network reconstructed with the nine transcription factor-encoding genes (the nodes with incoming interactions) differentially expressed in HPs derived from HE:DLL4$^+$. Size of the nodes represents relative abundance of mRNA of the respective gene, computed as log2(fold change) in DLL4$^+$ vs. DLL4$^−$ (see circle size scale below). Statistically insignificant changes in mRNA abundance (examples: GATA1, GATA2) were set to zero. Upregulation effects are mapped onto the node size as indicated. The color density represents enrichment of known targets of that transcription factor (regulon members) among the differentially expressed genes (see −log10(FDR) color scale below). Network visualization was performed using Cytoscape ver. 3.4.0[74]. **d** Schematic diagram of NOTCH regulation on HE specification and EHT. The most immature hPSC-derived CD144$^+$CD43$^−$CD73$^−$ HE cells expressing NOTCH1 but lacking arterial and venous identity arise on day 4 of differentiation. NOTCH activation induces specification of immature HE into DLL4$^+$ arterial-type HE, first detectable on day 5 of differentiation. Hematopoiesis from arterial-type HE:DLL4$^+$ is NOTCH dependent and produces definitive hematopoietic progenitors. Day 4 HE cells that are not DLL4$^+$ by day 5 of differentiation undergo EHT independent of NOTCH activation and produce NOTCH-independent hematopoietic progenitors with primitive potential

(Supplementary Fig. 7a). As shown in Supplementary Fig. 7b, c, siRNA-mediated SOX17 knockdown markedly reduced percentages and absolute numbers of T-cells produced from hESC-derived HPs, thereby providing direct evidence that arterial gene SOX17 is required for establishing definitive hematopoiesis.

RNAseq analysis of NOTCH ligands, receptors, and their downstream targets in D5 DLL4$^+$ and DLL4$^-$ HE, and HPs obtained from these populations, revealed D5 DLL4$^+$ AHE express greater levels of NOTCH1, NOTCH4, DLL4, and JAG2 as compared to DLL4$^-$ HE. However, expression of NOTCH-associated molecules and SOX17 was substantially lower in HPs, including HPs generated from DLL4$^+$ AHE on OP9-DLL4, suggesting a downregulation of NOTCH signaling and arterial program following EHT (Supplementary Fig. 8). These findings are consistent with observations in the mouse system which demonstrated that downregulation of NOTCH1 and SOX17 is essential for EHT[48]. The exact mechanism of NOTCH down-regulation at EHT stage remains unknown. Although NOTCH receptors are activated by cell-surface ligands in neighboring cells (trans-activation of NOTCH), NOTCH ligands expressed by the same cell typically inactivate NOTCH signaling (cis-inhibition of NOTCH)[49]. Although the response to trans-Delta is graded, cis-Delta response is abrupt and occurs at fixed threshold[50]. Thus, it is likely that in response to trans-DLL4 signaling from OP9-DLL4, AHE upregulates DLL4 expression to the threshold level required for cis-inhibition of NOTCH signaling in its own NOTCH1-expressing AHE cells, thereby allowing for EHT to proceed. This interpretation is consistent with studies in the mouse system, which have demonstrated that expression of NOTCH ligands, including DLL1 and DLL4, in the AGM vascular niche with co-expression of DLL4 and NOTCH1 on emerging hematopoietic cells is critical for HE to undergo EHT and subsequent HSC amplification through limiting NOTCH1 receptor activation by cis-inhibition[30,51]. However, despite downregulation of SOX17 and NOTCH1 expression following transition from DLL4$^+$ HE, we observed an enrichment of their known target genes (regulon members) in lin$^-$CD34$^+$CD45$^+$ progenitors at post-EHT stage in OP9-DLL4 cultures (Fig. 6c). These finding suggest that following EHT, the expression of arterial genes decreases, but downstream program activated by these genes in the presence of NOTCH ligands remains active.

Together, these results imply that arterial-type CD144$^+$ CD43$^-$CD73$^-$DLL4$^+$ HE represents the precursor of definitive NOTCH-dependent hematopoiesis with broad lympho-myeloid and definitive erythroid potential, while the CD144$^+$CD43$^-$ CD73$^-$DLL4$^-$ phenotype is associated with emerging immature HE endothelium (D4) or HE that has primitive NOTCH-independent hematopoietic potential (D5).

## Discussion

In the current study, we revealed that NOTCH signaling is essential for specification of definitive lympho-myeloid hematopoiesis by eliciting arterial specification of HE from hPSCs. We demonstrated that NOTCH activation promotes formation of transient CD144$^+$CD43$^-$CD73$^-$DLL4$^+$ HE population with high expression of arterial genes and active Runx1 + 23 enhancer that mark arterial-type HE in AGM, umbilical, and vitelline arteries[26,27,33,34,37]. Although CD144$^+$CD43$^-$CD73$^-$DLL4$^+$ AHE have lower hemogenic capacity compared to DLL4$^-$ HE, the hematopoietic potential of AHE is strictly NOTCH dependent. AHE is specified from CD144$^+$CD43$^-$CD73$^-$DLL4$^-$ immature HE cells emerging on D4 of differentiation in a NOTCH-dependent manner following acquisition of an arterial CD144$^+$CD43$^-$CD73$^-$DLL4$^+$ phenotype, while CD144$^+$ CD43$^-$CD73$^-$DLL4$^-$ HE cells that failed to undergo arterial

specification on day 5 of differentiation retained mostly primitive hematopoietic potential and were minimally affected by NOTCH activation (Fig. 6d). Demonstrating that definitive hematopoietic potential is highly enriched in arterial-type HE is in concordance with in vivo studies that established the restriction of lymphoid cell and HSC formation to the arterial vasculature in the yolk sac and embryo proper[5,13,14], enrichment of HSC precursors in DLL4$^+$ HE in AGM region[51], and selective impairment of AGM hematopoiesis in Efnb2$-/-$ mice[52]. Notably, DLL4$^+$ HE produced blood cells only on OP9-DLL4, but failed to undergo EHT in DLL1-Fc cultures in defined serum- and stroma-free conditions, suggesting that AHE, in contrast to non-AHE, requires some additional signaling factor, either soluble factors in serum, matrix proteins or a paracrine signaling between the OP9-DLL4 and AHE, that are necessary for EHT, but have yet to be determined.

Previous studies have found that NOTCH pathways are active during hematopoietic differentiation of mouse and hPSCs. The transient NOTCH activation increased the generation of CD45$^+$ HPs[25,53,54] and NOTCH inhibition with DAPT decreased the percentage of CD45$^+$ cells in cultures of hPSC-derived CD34$^+$CD73$^-$CD43$^-$ progenitors[21]. NOTCH activation in hPSC cultures is predominantly mediated through the NOTCH ligand, DLL4, expressed by endothelial cells[25]. However, since prior studies demonstrated that NOTCH signaling also contributes to expansion of embryonic and fetal HSPCs[30,31] and affects HSPC survival[55], it remains unclear whether DAPT treatment effects observed in prior PSC studies could be attributed to the NOTCH effect on EHT, post-EHT expansion or HP survival. It is also unknown, whether NOTCH signaling affects HE specification. In the present study, we provided evidence that NOTCH has several effects on hematopoiesis from HE. First, we demonstrated that NOTCH signaling is important for the specification of arterial-type HE cells with definitive hematopoietic program. In addition, NOTCH activation also potentiates the EHT from these cells, while having little effect on expansion and survival of blood cells at post-EHT stage.

Overall, our studies indicate that regulation of NOTCH signaling would be important to mimic the arterial HE, definitive lympho-myeloid hematopoiesis and HSC specification in hPSC culture. Nevertheless, we failed to achieve engraftment from NOTCH-activated cultures in our pilot studies. As previous studies have shown, the timing[56] and strength[57] of NOTCH signaling is tightly regulated during EHT and post-EHT. In mouse embryo, NOTCH signaling is especially critical for pre-HSC type I (VE-Cad$^+$ CD45$^-$CD41$^+$CD43$^+$) development and their transition into pre-HSC type II (Ve-Cad$^+$CD45$^+$CD41$^+$CD43$^+$) in the AGM, however later stages of HSC development are less dependent on NOTCH[58]. Thus, NOTCH signaling may need to be fine-tuned in hPSC cultures to establish long-term engrafting HSCs. It is also likely that acquisition of arterial features by HE is a necessary but not a sufficient prerequisite for HSC formation from hPSCs. In yolk sac, arterial vessels in contrast to venous vessels and capillaries produce lymphoid cells[13], yet arterial specification at extra-embryonic sites is not associated with establishing HSC program. Several signaling pathways, including those uncoupled from aortic specification such as HOXA[1,59,60], TGFβ[61], retinoic acid[59,62], inflammation[63,64], hormone[65,66], and blood-flow-induced shear stress[67,68], have all been shown to serve roles during HSC development and may be affected in in vitro generated cells. The investigation of the cross-talk among all of these different signaling pathways and manipulating them together is probably necessary to recapitulate embryonic definitive hematopoietic development and generate HSCs with robust and sustainable multilineage engraftment potential de novo.

## Methods

**hPSC maintenance and differentiation**. hPSCs, H1 and H9 hESC lines, DF19-9-7T fibroblast-hiPSC line, IISH2i-BM9 bone marrow-iPSC line, and IISH3i-CB6 cord blood-iPSC line (all from WiCell) were maintained and passaged in chemically defined conditions using vitronectin and E8 media and tested every 6 months for mycoplasma contamination. The hPSCs were differentiated into hema-toendothelial lineages using a chemically defined conditions[29]. On Day −1, hPSCs were singularized and plated on collagen IV-coated plates (0.5 µg/cm²) at a cell density of 7500 cells/cm² in E8 media supplemented with 10 uM Rock inhibitor (Y-27632, Cayman Chemicals). On Day 0, the media was changed to IF9S media supplemented with BMP4, FGF2 (50 ng ml⁻¹), Activin A (15 ng ml⁻¹, Peprotech), LiCl (2 mM, Sigma), and ROCK inhibitor (0.5 µM, Cayman Chemicals) and cultured in hypoxia (5% O₂, 5% CO₂). On day 2, the media was changed to IF9S media supplemented with FGF2, VEGF (50 ng ml⁻¹, Peprotech), and 2.5 µM TGFβ inhibitor (SB-431542, Cayman Chemicals). On day 4, cell cultures were singularized and stained with anti-CD31 microbeads (Miltenyi) for 15 min. The cells were washed and HE were purified using CD31 antibody and MACS LS columns (Miltenyi). Purified CD31⁺ HE were then plated at a density of 20,000 to 30,000 cells/cm² on collagen IV-coated plates (1 µg/cm²) that were either co-coated with IgG-Fc fragments or human DLL1-Fc (made in-house), in IF9S media supplemented with FGF2, VEGF, EGF, IGF-I, IGF-II, TPO, IL-6 (50 ng ml⁻¹), SCF (20 ng ml⁻¹), IL-3, FLT3L (10 ng ml⁻¹, Peprotech), and ROCK inhibitor (5 µM, Cayman Chemicals), and where specified, DMSO (1:1000, Fisher Scientific) or DAPT (10 µM, Cayman Chemicals), and cultured in normoxia (20% O₂, 5% CO₂). In some experiments, HE was cultured on plates co-coated with human JAG1-Fc (R&D Systems). A sample of the purified cells was analyzed by flow cytometry, and experiments were continued only if the purity of the HE was over 95% CD144⁺. On Day 4 + 1, the media was replaced with fresh media containing the same supplements without ROCK inhibitor. On day 4 + 3, extra media with the same supplements was added to the culture.

**OP9 maintenance and co-culture**. Mouse OP9 stromal cell line was kindly provided by Dr. Toru Nakano (Osaka University, Osaka, Japan). OP9, OP9-DLL4, and the inducible OP9-iDLL4 (made in-house) cell lines were maintained in αMEM with 20% FBS (GE) on gelatin-coated plates in normoxia. Using TrypLE (Thermo), OP9 were passaged at a 1:8 ratio every 3–4 days when they were 80% confluent. One day before co-culture with differentiated human HE cells, OP9 lines were treated with mitomycin C (1 mg ml⁻¹) for 2 h and then plated at a density of 12,500 cells/cm². D4 HE cells or D5 CD144⁺ subsets were plated onto OP9 lines at a density between 1000 and 2000 cells/cm² in media containing αMEM with 10% FBS (GE), TPO, SCF, IL-6 (50 ng/ml), IL-3, and FLT3L (10 ng/ml). Media was changed after 24 h, and extra media added 2 days later. Experiments conducted with DAPT were treated with 20 µM, while corresponding control conditions had DMSO added at a 1:500 dilution.

**Generation of OP9-DLL4, OP9-JAG1, and DOX-inducible OP9-iDLL4**. Human DLL4 gene fragment was amplified by PCR from a vector previously used to establish the OP9-DLL4 cell line, and the JAG1 gene was amplified by PCR from cDNA of D5 differentiation cultures that were treated with Sonic Hedgehog from D2–5, which has been found to increase Jag1 expression. The DLL4 and JAG1 gene fragment was subsequently cloned into a pSIN-EF1a-DLL4⁻IRES-Puro and pSIN-EF1a-JAG1-IRES-Puro lentiviral expression vector for the constitutively expressed OP9- DLL4 and JAG1 lines, respectively. Virus production and concentration was carried out by calcium phosphate transfection of Lenti-X 293T-cells (Clonetech, Mountain View, CA). After 12 h, virus-containing medium was replaced with fresh OP9 culture medium. After 3 days, the cells were treated with Puromycin for 2 weeks. For dox-inducible OP9-DLL4, the DLL4 gene fragment was subsequently cloned into a pPB-TRE-DLL4⁻P2A-Venus-EF1α-Zeo||EF1a-M2rtTA-T2A-Puro PiggyBac vector made in-house. OP9 cells were then transfected with pPB vector. Three days later the transfected OP9 cells were treated with Puromycin/Zeocin for 2 weeks. Samples of the OP9-iDLL4 cells were treated with doxycycline for 24 h, then DLL4 and Venus expression were confirmed by flow cytometry.

**Single-cell deposition assay for EHT**. One day before single-cell deposition, the OP9-iDLL4 cell line was treated with mitomycin C as described above, and passaged into 96-well plates at a density of 12,500 cells/cm². OP9-iDLL4 used for the NOTCH activation condition was incubated with doxycycline for 24 h after passaging into 96-well plates. On the day of single-cell sorting, OP9-iDLL4 media was changed to αMEM with 10% FBS (GE), TPO, SCF, IL-6 (50 ng ml⁻¹), IL-3, FLT3L (10 ng ml⁻¹), and DMSO (1:500) for the control, and NOTCH activation conditions, or DAPT (20 µM) for the NOTCH inhibition condition. Day 4 differentiated hPSCs were singularized, stained for CD309-PE and CD144-APC (Miltenyi Biotech), and was single-cell sorted into individual wells of the 96-well plates using a FACS Aria II. To exclude possibility of doublets, we used a low density (less than 1 million cells/ml) cells suspension, sorting speed less than 1000 cellular events/per second and stringent gating on single cells using both FSC-A vs. FSC-H and SSC-A vs. SSC-H. One day after sorting, the media was changed to fresh media without DMSO or DAPT, and extra media was added every 3 days. Seven days later, the plates were fixed and stained for immunofluorescent staining with anti-CD144 (rabbit, eBioscience) and anti-CD43 (mouse, BD Biosciences) primary

antibodies and anti-rabbit AlexaFluor488 and anti-mouse AlexaFluor594 secondary antibodies (Jackson Immunology) in order to score the hematopoietic/endothelial colonies.

**CellTracer proliferation assay and cell-cycle analysis**. D4 CD31⁺ HE cells were incubated in PBS with CellTracer (1 µg ml⁻¹, Thermo) for 20 min at 37 °C. After washing, the cells were plated on collagen IV-coated plates with either Fc-IgG or DLL1-Fc and the modified Day 4 media, as described above, at a higher density of 30,000–40,000 cells/cm² due to toxicity from the CellTracer. Aliquots of the purified cells were analyzed by flow cytometry to determine the purity of the MACS cells and establish the Generation 0 peak for the proliferation assay. Secondary cultures were collected every day after plating for flow cytometry analysis, and calibration beads were used to generate compatible CellTracer results. After D4 + 4, FlowJo Analysis software was used to concatenate the data from each day. The average number of cell divisions was calculated based on the number of cells on each day (Fig. 1f) and applied to the proliferation platform algorithm in Flowjo to determine the specific generation gates. Those peaks were re-applied to individual sets of data to determine the percentage of each generation within the hema-toendothelial populations. Apoptosis was detected used annexin V flow cytometry. Fluorescent reagents used for analysis, cell viability, apoptosis, and proliferation are listed in Supplementary Table 1. For cell-cycle analysis, D4 + 4 cells were incubated in culture media with EdU (10 µM, Thermo Fisher) for 2 h and stained with CD43 and CD144 antibodies for 20 min. For EdU detection, the Click-IT EdU Alexa Flour 647 kit (Thermo Fisher) with DAPI (4 µg ml⁻¹, Sigma) was used.

**T-Cell differentiation and T-cell LDA**. Total D4 + 4 cultures were singularized, strained, and cultured in T-cell differentiation conditions on OP9-DLL4 for 3 weeks[29]. For D5 + 4 cultures, only the floating hematopoietic cells were collected and cultured in T-cell differentiation conditions. LDAs were conducted with the floating cells collected from D5 + 4 cultures (HE:DLL4⁻ on OP9 + DAPT, OP9 + DMSO, and OP9-DLL4, and HE:DLL4⁺ on OP9-DLL4). Row A of a 96-well plate received 500 cells/well, and each subsequent row afterwards had half the previous row (Row B contained 250, Row C contained 125… Row H contained 3–4 cells). The wells were scored 2 weeks later by eye and flow-cytometry for CD5⁺CD7⁺ containing cells. Positive threshold was set at 167 CD5⁺CD7⁺ cells/well. Extreme limiting dilution analysis was conducted using the previously established algorithm[69].

**Red blood cell differentiation and maturation**. In order to assess the definitive erythropoietic potential of HP cells, we adopted our previously describe red blood cell differentiation protocol[36] to become chemically defined and feeder- and serum-free. Floating cells were collected, washed, and plated back into their respective cultures for D4 + 5 cells, or plated onto collagen IV-coated plates for D5 + 4 cells, with IF9S supplemented with dexamethasone (10 µM), EPO (2 U ml⁻¹), SCF, FLT3L, TPO, IL-6 (100 ng ml⁻¹), and IL-3 (10 ng ml⁻¹). Extra media with the same supplements was added 2 days later. An additional 2 days later, the cultures were treated with half-media changes every 2 days with IF9S supplemented with dexamethasone (10 µM), SCF (100 ng ml⁻¹), and EPO (2 U ml⁻¹). The floating cells were collected 10 days later to analyze by flow cytometry and RNA isolated for qPCR analysis.

**Generating Runx1 + 23 enhancer reporter hESC line**. Runx1 + 23 enhancer fragment[27] was amplified by PCR and subsequently cloned into the AAVS1-SA-2A-PURO vector (gift from Gadue Lab, The Children's Hospital of Philadelphia). Human ESCs were transfected with zinc-finger nuclease vectors and later puromycin-resistant individual cells were clonally expanded and on-targeted clones were selected. Southern Blot (SB) analysis was performed by DIG-labeling hybridization (Roche). Briefly, 10 µg genomic DNA was digested using a EcoRV restriction enzyme for overnight, separated on a 0.7% agarose gel for 6 h, transferred to a nylon membrane (Amersham), and incubated with DIG-labeling probes. The external probe is a DIG-labeled 600 nucleotide fragment that binds to the EcoRV-digested fragment of the 5′ external region. The internal probe is a DIG-labled 700 nucleotide fragment that binds to the EcoRV-digested fragment of the of the eGFP region.

**SOX17 knockdown in differentiation cultures using siRNA**. hESCs were transfected with 50 nM SOX17 siRNA SMARTpool (Dharmacon, GE) or 50 nM scramble negative control siRNA (Dharmacon) on D3 of differentiation using Lipofectamine 3000 (Invitrogen). To confirm SOX17 knockdown, SOX17 mRNA levels were assessed 48 h after transfection with qPCR. HPs from control and SOX17 siRNA treated cultures were collected on day 8 of differentiation and assessed for T-cell potential as described above.

**Hematopoietic colony forming unit assay**. Hematopoietic colony forming unit assay was conducted in serum-containing H4436 Methocult (Stem Cell Technologies) according to manufacturer instruction.

**Flow cytometry and FACS-sorting**. Flow cytometry was conducted using the MACSQuant Analyzer 10 (Miltenyi Biotech). FACS-sorting was conducted on a FACS Aria II (BD). Antibodies used for flow cytometic analysis and cell sorting are listed in Supplementary Table 2.

**Western blot**. Cell extracts were prepared by adding IP Lysis buffer (Thermo Scientific) with protease inhibitor cocktail (Sigma). Cell lysates (10 μg) were separated by 6% SDS-PAGE. Separated proteins were transferred to a PVDF membrane, and were stained with Notch1, Notch1-ICD antibody (Cell Signaling Technology), or Actin (Santa Cruz). Immunoblots were visualized using the ECL PLUS detection kit (Amersham Pharmacia). Quantitative analysis of protein expression was performed using Image Lab software version 5.2 (Bio-Rad). Antibody dilutions are presented in Supplementary Table 2. All uncropped western blots can be found in Supplementary Fig. 9.

**qPCR analysis**. Cells were differentiated for the respective days and sorted on a FACS Aria II. RNA was collected using RNA MiniPrep Plus (Invitrogen) and quantified on a NanoDrop (GE Healthcare). Equal amounts of RNA were used for cDNA synthesis using SuperScript III First-Strand Synthesis System (Life Technologies). qPCR was conducted using Platinum SYBR Green qPCR SuperMix (Life Technologies). The reactions were run on a Mastercycler RealPlex Thermal Cycler (Eppendorf) and the expression levels were calculated by minimal cycle threshold values (Ct) normalized to the reference expression of RPL13a. The qPCR products were run on an agarose gel and stained with ethidium bromide to confirm specificity of the primers. Primer sequences can be found in Supplementary Table 3.

**RNA-Seq data processing and analysis**. Total RNA was isolated from the D4 HE, D5 HE:DLL4$^+$ and HE:DLL4$^-$ and CD235a/CD41a$^-$CD34$^+$CD45$^+$ derived from HE:DLL4$^+$ and HE:DLL4$^-$ cells using the RNeasy mini Plus Kit (Qiagen). RNA purity and integrity was evaluated by capillary electrophoresis on the Bioanalyzer 2100 (Agilent Technologies, Santa Clara, CA). One hundred nanograms of total RNA was used to prepare sequencing libraries using the TruSeq RNA Sample Preparation kit (Illumina, San Diego, CA). Final cDNA libraries were quantitated with the Qubit Fluorometer (Life Technologies, Carlsbad, CA) and multiplexed with eighteen total indexed libraries per lane. Sequencing was performed using the HiSeq 3000 (Illumina, San Diego, CA) with a single read of 64 bp and index read of 7 bp.

Base-calling and demultiplexing were completed with the Illumina Genome Analyzer Casava Software, version 1.8.2. Following quality assessment and filtering for adapter molecules and other sequencing artifacts, the remaining sequencing reads were aligned to transcript sequences corresponding to hg19 human genome annotation. Bowtie v 1.1.2 was used, allowing two mismatches in a 25 bp seed, and excluding reads with more than 200 alignments[70]. RSEM v 1.3.0 was used to estimate isoform or gene relative expression levels in units of 'transcripts per million' (tpm), as well as posterior mean estimate of the 'expected counts' (the non-normalized absolute number of reads assigned by RSEM to each isoform/gene)[71]. R statistical environment (R core team, 2014) was used at all of the stages of downstream data analysis. The entire set of libraries was pre-normalized as a pool using median normalization routine from EBSeq package[72]. EBSeq with 10 iterations was applied to call for differential expression. The EBSeq's default procedure of filtering low-expressed genes was suppressed by setting the QtrmCut parameter to zero. Genes with assigned value of Posterior Probability of Differential Expression above 0.95 were preliminary selected. Subsequently, only genes demonstrating the Critical Coefficent[73] value above 1.5 were retained as differentially expressed.

**Statistical analysis**. Sample sizes were determined by the nature of the experiment and variability of the output, not by a statistical method. The experiments were not randomized and the investigators were not blinded to allocation during experiments and outcome assessment. Statistical analysis was performed in PRISM software. Data obtained from multiple experiments were reported as mean ±standard error. Where appropriate, either a one-way ANOVA or two-way ANOVA were utilized with a Bonferroni post-hoc test. Differences were considered significant when $*p < 0.05$, $**p < 0.01$, or $***p < 0.001$.

**Data availability**. The authors declare that all data supporting the findings of this study are available within the article and its Supplementary Information Files or from the corresponding author upon reasonable request. RNAseq data have been deposited in the NCBI Gene Expression Omnibus (GEO) database under accession codes GSE95028 and GSE96815.

https://www.ncbi.nlm.nih.gov/geo/query/acc.cgi?acc=GSE95028
https://www.ncbi.nlm.nih.gov/geo/query/=GSE96815

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

## Acknowledgements

We thank Dr. Toru Nakano (Osaka University, Osaka, Japan) for providing OP9 cells and Dr. Paul Gadue (Children's Hospital of Philadelphia, Philadelphia, PA) for providing AAVS1 targeting construct. This work was supported by funds from the National Institute of Health (R01HL116221, U01HL099773, P51 RR000167, U01HL10001, and UO1 HL100395) and The Charlotte Geyer Foundation.

## Author contributions

G.I.U. designed, conducted, and analyzed experiments; interpreted experimental data; made figures; and contributed to the concept and wrote manuscript. H.S.J. designed, conducted, and analyzed experiments; interpreted experimental data; made figures; generated and characterized the Runx1+ 23 WA01 hESC line; and contributed to writing of the paper. A. K. performed lymphoid differentiation assay. M.A.P. generated the inducible OP9-iDLL4 cell line. B.K.H. provided the DLL1-Fc ligand and advised on experimental design. C.E.Z. and D.J.T. generated the OP9-JAG1 cell line and conducted OP9-JAG1 experiments. E.M. and M.R. conducted and analyzed experiments. O.M. and S.S. performed bioinformatics analysis of RNA-seq studies. O.J.T. and L.Z. provided Runx1+ 23 reporter construct and advised on experimental design. J.A.T. directed the RNA-seq studies. I.D.B. provided DLL1-Fc ligand and advised on experimental design. I.I.S. developed the concept, led and supervised the studies, analyzed and interpreted data, and wrote the manuscript.

## Additional information

**Competing interests:** The authors declare no competing interests.

