## [Peer Review File · Nature Communications]

Reviewers' Comments:

Reviewer #1:

Remarks to the Author:

The paper by Uenishi et al. studies an important issue in hematopoiesis from hemogenic endothelium. It showed that Notch signaling plays a critical role in driving arterial specification in hemogenic endothelium from pluripotent stem cells. Although the role of notch in this process has been suggested before, there has not been clear scientific evidence to demonstrate this. This manuscript fills this knowledge gap.

Although manuscript is well-written, there are a few issues with data quality that need to be addressed.

1. In Figure 2d, the table showing results for the single cell deposition assay indicates that 288 cells were used; however, the three types of colonies account for less than 40% of total. Please explain what is the fate of the remaining 60%? are cell dying or are they quiescent or are they senescent?

2. Figure 2d showed a mixed colony and in table this outcome is rare. What measures did authors take to assure this is not due to doublets? Please explain in methods.

3. Figure 2e for CellTracer experiments show generation gates. Please add a positive control or sample used to define these gates? otherwise please explain gating strategy on methods.

4. Figure 3c shows T cell differentiation potential of HPs. This dot plots showing flow cytometry do not meet minimal standards. Quadrants show percentages while total of events is clearly less than 100 events. Please show more events in dot plot.

5. The finding that the feeder-free DLL1-Fc treatment of hPSC-HE did not give rise to blood cells but it did when plated in OP9-DLL4 is very interesting and requires some explanation in the discussion.

Reviewer #2:

Remarks to the Author:

The authors investigate the role of Notch signaling in the specification of "arterial-type" hemogenic endothelial cells from hPSC, and the hematopoietic potential of hemogenic endothelial cell subtypes, based on Dll4 expression. There are a number of concerns with this paper.

1. Although these findings advance the understanding and manipulation of this culture system, the findings are not particularly novel. For example, a role for Notch signaling in hemogenic endothelial cell specification has been shown several years ago (Marcelo et al., Dev Cell 2013). Also, the role of Notch in all aspects of EHT is known to be complex (i.e. Butko et al., Dev Biol 2016), and there are no new mechanistic data to show how Notch signaling is actually exerting different effects on different cell types (i.e. "specification" of arterial-type hemogenic endothelial cells vs. potential of HSPC) during the progression of hematopoiesis in this culture system.

2. There is no direct evidence that arterial gene expression is required for definitive hemogenic endothelial cell specification or function. If arterial gene expression is selectively inhibited, does this change hematopoietic potential?

3. In fact, in the AGM in vivo, the aortic endothelial cells are "arterial" by definition before they become hematopoietic, and the arterial program has to be downregulated (Lizama et al., Nature Comm 2015) to enable hematopoietic function.

4. On a related note, it is known, and shown here, that Notch increases arterial gene expression in endothelial cells, but only a very small subset of arterial endothelial cells acquire blood-forming potential. Notch also regulates lymphatic endothelial cell development, and these are not known to be blood-forming. So, clearly Notch is involved, but probably not the critical step in the process.

5. The authors isolate the CD144+CD43-CD73- population based on magnetic sorting for CD31; it should be shown that these populations are the same, and there are no CD73+ cells in this group, given they suggest that this distinguishes "hemogenic" endothelial cells.

6. The authors state that all definitive hemogenic endothelial cells are derived from arterial endothelium, but that is not correct. In the murine yolk sac, for example, definitive hemogenic endothelial cells arise from primitive endothelial cells, simultaneously with arterial and venous endothelial cells during vascular remodeling.

7. The graphs are difficult to read and it would be helpful if the controls were shown first, as the left-most bars, in a consistent manner; currently, the controls are displayed differently in different figures.

Reviewer #3:

Remarks to the Author:

In the ms by Uenishi et al., the authors address the mechanisms underpinning early aspects of hematopoietic differentiation, with a focus on definitive hematopoiesis and the hemogenic endothelium. The authors provide data for a role of Notch signaling to i) activate definitive hematopoiesis and ii) in the endothelium-to-hematopoietic transition resulting in a broad spectrum of differentiated phenotypes.

The manuscript is not the first to invoke a role for Notch in the early phases of hematopoietic differentiation, but it provides important information about the role of Notch in distinct steps in the process, and also provides a working protocol which provides a larger array of differentiated cells, encompassing both lympho-myeloid as well as erythroid cells, which extends the previously published protocol (ref 35). The data also shed light on the ongoing discussion whether the arterial compartment of the early vasculature provides a niche or is de facto required for initiating the program for definitive hematopoiesis.

Major critique:

1. The authors put a strong focus on NOTCH1 and the DLL4 ligands. In Figure 1, they report that NOTCH1 comes on at day 4 and DLL4 the day after. For completion, to make this claim, it would be good to establish that these are the only receptors and ligands expressed at these stages (it could well be that they are).

2. The authors use immobilized Dll1 to activate Notch and a gamma-secretase inhibitor (DAPT) to block Notch. These are well established tools and I have in principle no problem to this as gain- or loss-of-function approaches. DAPT, depending on dosing, however can negatively impact cells, and although the authors do not see enhanced apoptosis (line 186 ->), it would be good to know how the concentration used was determined (and where toxicity kicks in at higher concentrations). As regards Dll1 as an inducer, it would be interesting to see if also ligands on the Jagged1 side would be competent in activation. Fc-Jag1 is easily available and could be tested. This has bearings on trans-activation and cis-inhibition, and the possible roles of Fringe, as discussed below.

3. For the experiments in Figure 3 (line 206 ->), the authors seeded cells in methocellulose to assess colony formation. It is unclear how an immobilized Notch ligand would gain access to the cells in this assay (it is tricky enough with gelatin in 2D culture). The authors need to comment on this and also provide good evidence that Notch signaling is activated in the cells in this assay, for

example by using a fluorescence or luciferase-based Notch reporter.

4. The observation in Figure 5d (line 290->) that DAPT affects hematopoietic activity in the Dll4^{high}, but not in the Dll4^{low} population is intriguing. It is surprising that the cells with high levels of ligand would be most responsive, given that high levels of ligand in a cell can lead to cis-inhibition. The authors need to comment on this observation.

5. The model in the summary figure (Figure 6D) is intriguing, but the subdivision into a Notch1 low, Dll4⁻ (Notch-independent) and a Notch1 High Dll4⁺ (Notch-dependent) population poses a problem from a Notch-signaling perspective, namely who signals to who (see also comment 4 above)? In many other situations cells with high level of ligand are juxtaposed with cells expressing high Notch receptor but low ligand levels. The authors need to carefully think about the consequences of their model with regard to trans-activation vs cis-inhibition and whether other ligands or Fringe genes are expressed. A particularly interesting case of Notch signaling involving more than one ligand type and Fringes is the tip/stalk cells in the growing endothelium.

6. What happens if OPA-Dll4 cells are replaced with immobilized ligand (Dll or Jag) in the experiments where differentiation outcomes are assessed (Figure 2D, line 160 ->)?

Minor comments:

1. The Abstract is difficult to grasp, in particular for a more broad audience in Nature Communications. The first few sentences are somewhat abstract and in fact a bit pompous; after reading them one expects all major issues in hematopoiesis to be solved. I would favor toning in tone a bit, shorten the general part and more specifically phrase the question "in a layman's terms".

2. Discussion, line 355: In current study -> In the current study....

3. For Fig 1C, I could not find the information about the antibody used to detect activated Notch, is it a VAL-1744 ab?

RESPONSE TO REVIEWERS NCOMMS-17-08395A-Z.

We greatly appreciate the thoughtful reviews. In the revised manuscript, we completed all studies requested by the reviewers and made corresponding modifications in the text indicated by red font. Please see reply to reviewer's comments below.

As requested by reviewers, we included a new RNAseq data and provided an additional GEO number related to the second set of data. In addition, we added two new co-authors who was involved in JAG1 experiments requested by reviewers and scale definition to Fig.6. We also corrected sequence of figures in Supplement to ensure that all figures are cited in order.

Reviewer #1 (Remarks to the Author):

The paper by Uenishi et al. studies an important issue in hematopoiesis from hemogenic endothelium. It showed that Notch signaling plays a critical role in driving arterial specification in hemogenic endothelium from pluripotent stem cells. Although the role of notch in this process has been suggested before, there has not been clear scientific evidence to demonstrate this. This manuscript fills this knowledge gap.

Although manuscript is well-written, there are a few issues with data quality that need to be addressed.

1. In Figure 2d, the table showing results for the single cell deposition assay indicates that 288 cells were used; however, the three types of colonies account for less than 40% of total. Please explain what is the fate of the remaining 60%? are cell dying or are they quiescent or are they senescent?

Response: This is a technical issue. Viability after single cell sorting of endothelial cells is commonly very low compared to other cell types (around 50% to 70%). In addition, single cell sorting into 96-well plates has some inherent error, as some droplets do not hit the center of the well. However, while the three types of colonies account for less than 40% of the total, the fact that the total number of colonies is consistent across each of the 3 NOTCH conditions suggests the sorting conditions were consistent. Vulnerability of HE generated in hPSC cultures was also recognized by Gordon Keller (Ditadi et al, 2015) and George Daley groups (Sugimura et al, 2017).

2. Figure 2d showed a mixed colony and in table this outcome is rare. What measures did authors take to assure this is not due to doublets? Please explain in methods.

Response: The sorting conditions set up on the FACSaria were extremely stringent to ensure no doublets were sorted into a single well. Both FSC-A vs FSC-H and SSC-A vs SSC-H were utilized to ensure all doublets were aborted. In addition, we used a low density (less than 1 million cells/ml) cells suspension and sorting speed less than 1000 cellular

events/per second. Typically, with these precautions, doublet formation should be negligible. We expanded materials and method section to describe a single cell sorting procedure in details on pages 22-23.

3. Figure 2e for CellTracer experiments show generation gates. Please add a positive control or sample used to define these gates? otherwise please explain gating strategy on methods.

Response: Gating strategy was based on an algorithm provided by FlowJo's proliferation assay. Based on the number of cells plated to the number of cells collected (Figure 1F), we determined the average number of cell divisions to be 7, and the gates were determined by concatenating the data. We expanded gating strategy in the methods section (page 23) and added supplemental Figure S2a to show gates for d4+4 cells.

4. Figure 3c shows T cell differentiation potential of HPs. This dot plots showing flow cytometry do not meet minimal standards. Quadrants show percentages while total of events is clearly less than 100 events. Please show more events in dot plot.

Response: The reason there are very low event numbers in Figure 3c, +DAPT condition is because there were little to no T-cell progenitors, which is the result we are trying to show. In the bar graphs below, we illustrate that owing to the fact that there are no human CD45+ cells, hence the low number of events in the flow plot. In contrast, HE cultured on DLL1-Fc produced high number of T cells (Fig. 1c right plot showing over 1000 events when HE cultured on DLL1-Fc). Below is a plot showing at least 2500 events from our limiting dilution assay, proving that we can get sufficient CD4+CD8+ T cell progenitors from HE:DLL4+ derived HPs.

5. The finding that the feeder-free DLL1-Fc treatment of hPSC-HE did not give rise to blood cells but it did when plated in OP9-DLL4 is very interesting and requires some explanation in the discussion.

Response: This is specific to D5 HE:DLL4+, the arterial-type HE (AHE). Our feeder-free conditions are very minimalistic, and there may be some signaling factor, either through

soluble factors in serum, matrix proteins or a paracrine signaling factor between the OP9-DLL4 and AHE, that are necessary for EHT that have yet to be determined. as suggested we added this explanation in discussion on page 17.

Reviewer #2 (Remarks to the Author):

The authors investigate the role of Notch signaling in the specification of “arterial-type” hemogenic endothelial cells from hPSC, and the hematopoietic potential of hemogenic endothelial cell subtypes, based on DLL4 expression. There are a number of concerns with this paper.

1. Although these findings advance the understanding and manipulation of this culture system, the findings are not particularly novel. For example, a role for Notch signaling in hemogenic endothelial cell specification has been shown several years ago (Marcelo et al., Dev Cell 2013). Also, the role of Notch in all aspects of EHT is known to be complex (i.e. Butko et al., Dev Biol 2016), and there are no new mechanistic data to show how Notch signaling is actually exerting different effects on different cell types (i.e. “specification” of arterial-type hemogenic endothelial cells vs. potential of HSPC) during the progression of hematopoiesis in this culture system.

Response: In this manuscript, we demonstrated for the first time that 1) NOTCH signaling is required for arterial specification of HE, 2) identified phenotype of arterial HE vs non-arterial-HE; 3) demonstrated direct progenitor-progeny link between arterial type HE and definitive hematopoiesis; 4) and provided direct evidence that NOTCH signaling regulates EHT. These findings have important implications in our understanding of hematopoietic development and further advancing hPSC technologies for the production of blood cells for transplantation in immunotherapy.

As we discussed in introduction, although shared requirements for Notch, VEGF, and Hedgehog signaling for both arterial fate acquisition and HSC development, led to the hypothesis that arterial specification could be a critical prerequisite for HSC formation, a direct progenitor-progeny link between arterial of progenitor and definitive hematopoiesis has never been demonstrated. Moreover, several studies (including the most recent study by Ditadi et al, 2015) concluded that arterial fate and definitive blood development are uncoupled processes. Thus, it was very critical to prove or disprove a link between arterial programming of HE and definitive hematopoiesis.

In addition, we examined NOTCH effect on blood cells at post EHT stage (Please see “NOTCH Activation Maintains Multilineage Potential and Increases Definitive Characteristics of Hematopoietic Progenitors Emerging from HE” section of manuscript). These studies showed the importance of NOTCH signaling for the maintenance of the multilineage potential of hematopoietic progenitors emerging from HE.

Marcelo et al, 2013 studied the effect of NOTCH inhibition with DAPT on generation of Flk1+c-kit+ cells in whole embryo cultures. Flk1+c-kit+ typically marks rounding cells at advanced stages of EHT in AGM (Dzierzak, Speck, Medvinsky, Bruijn and other groups). We studied a NOTCH effect on HE with epithelioid morphology, i.e. before EHT, which is typically identified by c-kit-negative Runx1-reporter and/or Ly6a-reporter-positive phenotype in mouse system. Moreover, neither Marcelo nor other groups revealed phenotypic features of arterial type HE versus non-arterial HE or demonstrated the differences in hematopoietic potential and response to NOTCH signaling between DLL4+ arterial HE and DLL4- non-arterial HE. Whether NOTCH is required for acquisition of arterial characteristics by epithelioid HE cells before EHT was not addressed in prior studies as well.

2. There is no direct evidence that arterial gene expression is required for definitive hemogenic endothelial cell specification or function. If arterial gene expression is selectively inhibited, does this change hematopoietic potential?

Response: The main goal of this study is to demonstrate the existence of arterial HE and direct progenitor-progeny link between arterial precursor and definitive hematopoiesis. We are currently exploring how modulation of arterial genes affect definitive hematopoiesis. These findings will be reported in a separate manuscript.

3. In fact, in the AGM in vivo, the aortic endothelial cells are “arterial” by definition before they become hematopoietic, and the arterial program has to be downregulated (Lizama et al., Nature Comm 2015) to enable hematopoietic function.

Response: The studies by Lizama et al, 2015 demonstrated that following EHT, hematopoietic cells lose expression of NOTCH1 and SOX17 arterial genes. Similarly, we observed a significant reduction in expression of these two arterial genes in hematopoietic progenitors following EHT from DLL4+ HE even in OP9-DLL4 cocultures, as compared to their DLL4+ arterial HE ancestors (please see added supplemental Fig.S7). However, we observed an enrichment of known targets (regulon members) of SOX17 and NOTCH1 in lin-CD34+CD45+ progenitors emerging from DLL4+ HE (Fig. 7c). These findings suggest that following EHT, the expression of arterial genes decreases, but downstream program activated by these genes remains active. Please see added discussion on page 16.

4. On a related note, it is known, and shown here, that Notch increases arterial gene expression in endothelial cells, but only a very small subset of arterial endothelial cells acquire blood-forming potential. Notch also regulates lymphatic endothelial cell development, and these are not known to be blood-forming. So, clearly Notch is involved, but probably not the critical step in the process.

Response: It is known that NOTCH increases expression of arterial genes in “generic” endothelium. However, it has become increasingly recognized that HE is a distinct lineage of vascular endothelium, development of which is regulated by distinct transcriptional program. Here we demonstrated for the first time that NOTCH signaling induces arterialization of HE. AHE comprises a small proportion of cells in hPSC cultures. In the AGM, a very small subset of hemogenic endothelium give rise to definitive HSCs.

We agree that NOTCH signaling regulates lymphatic development, however analysis of RNAseq data reveals very low expression of lymphatic markers PROX1 and LYVE1 in HE at all stages of development.

5. The authors isolate the CD144+CD43-CD73- population based on magnetic sorting for CD31; it should be shown that these populations are the same, and there are no CD73+ cells in this group, given they suggest that this distinguishes “hemogenic” endothelial cells.

Response: This has already been shown in our previous publications (Choi et al, 2012 and Uenishi et al, 2014). We also added supplementary Figure S1a to demonstrate lack of CD73 and CD43 expression in day 4 HE.

6. The authors state that all definitive hemogenic endothelial cells are derived from arterial endothelium, but that is not correct. In the murine yolk sac, for example, definitive hemogenic endothelial cells arise from primitive endothelial cells, simultaneously with arterial and venous endothelial cells during vascular remodeling.

Response: As we discuss in the introduction, thorough analysis of yolk sac hematopoiesis by Nancy Speck’s group (Yzaguirre, A.D. & Speck, N.A., 2016) revealed that definitive EMPs arises from venous and arterial vessels, however lymphoid development even in yolk sac is restricted to arterial vessels. These findings are in concordance with our observation of direct link between the arterial program and lymphopoiesis.

7. The graphs are difficult to read and it would be helpful if the controls were shown first, as the left-most bars, in a consistent manner; currently, the controls are displayed differently in different figures.

Response: We’ve positioned the “Controls” in the center in all conditions consistent with the assumed level of NOTCH signaling in the experiments from low (+DAPT), mid (Control), and high (+DLL1-Fc) NOTCH activation. Figure 5 Controls are positioned differently because they are the unmanipulated conditions.

Reviewer #3 (Remarks to the Author):

In the ms by Uenishi et al., the authors address the mechanisms underpinning early

aspects of hematopoietic differentiation, with a focus on definitive hematopoieses and the hemogenic endothelium. The authors provide data for a role of Notch signaling to i) activate definitive hematopoiesis and ii) in the endothelium-to-hematopoietic transition resulting in a broad spectrum of differentiated phenotypes.

The manuscript is not the first to invoke a role for Notch in the early phases of hematopoietic differentiation, but it provides important information about the role of Notch in distinct steps in the process, and also provides a working protocol which provides a larger array of differentiated cells, encompassing both lympho-myeloid as well as erythroid cells, which extends the previously published protocol (ref 35). The data also shed light on the ongoing discussion whether the arterial compartment of the early vasculature provides a niche or is de facto required for initiating the program for definitive hematopoiesis.

Major critique:

1. The authors put a strong focus on NOTCH1 and the DLL4 ligands. In Figure 1, they report that NOTCH1 comes on at day 4 and DLL4 the day after. For completion, to make this claim, it would be good to establish that these are the only receptors and ligands expressed at these stages (it could well be that they are).

Response: We added a heatmap to demonstrate expression of NOTCH ligands, receptors, and their downstream targets and arterial genes on different stages of development (Supplementary Figure S7). Discussion of these findings is presented on page 16.

2. The authors use immobilized Dll1 to activate Notch and a gamma-secretase inhibitor (DAPT) to block Notch. These are well established tools and I have in principle no problem to this as gain- or loss-of-function approaches. DAPT, depending on dosing, however can negatively impact cells, and although the authors do not see enhanced apoptosis (line 186 ->), it would be good to know how the concentration used was determined (and where toxicity kicks in at higher concentrations). As regards Dll1 as an inducer, it would be interesting to see if also ligands on the Jagged1 side would be competent in activation. Fc-Jag1 is easily available and could be tested. This has bearings on trans-activation and cis-inhibition, and the possible roles of Fringe, as discussed below.

Response: As for DAPT concentrations, we conducted experiments to determine what concentrations are below the toxicity levels. After we determined the maximum dosage, we conducted a western blot to ascertain that NICD was inhibited. This is how we determined that 5uM was sufficient for serum/feeder-free conditions, but 20uM of DAPT was necessary for OP9 conditions. We used DLL1-Fc as an inducer, because we found a significant effect of DLL1-Fc on HE development. While we did test JAG-Fc and OP9-JAG1, we did not see a significant change in hematopoietic activity between controls versus JAG1 cultures. We added figures S1d and S1f in the Supplementary file to illustrate the lack of JAG1 effect on HE.

3. For the experiments in Figure 3 (line 206 ->), the authors seeded cells in

methocellulose to assess colony formation. It is unclear how an immobilized Notch ligand would gain access to the cells in this assay (it is tricky enough with gelatin in 2D culture). The authors need to comment on this and also provide good evidence that Notch signaling is activated in the cells in this assay, for example by using a fluorescence or luciferase-based Notch reporter.

Response: In Figure 3a, we evaluated CFC potential of blood progenitors that were derived from HE cultured in presence of DAPT, DLL1-Fc, or DMSO (control). NOTCH was not manipulated in the CFU assay. We modified text on page 9 and Figure 3a legend to avoid confusion.

4. The observation in Figure 5d (line 290->) that DAPT affects hematopoietic activity in the Dll4^{high}, but not in the Dll4^{low} population is intriguing. It is surprising that the cells with high levels of ligand would be most responsive, given that high levels of ligand in a cell can lead to cis-inhibition. The authors need to comment on this observation.

Response: The reviewer raises a very interesting question. As we discussed in response to the second reviewer, NOTCH signaling decreases following EHT. It is possible that high levels of ligand at EHT initiation may be required for cis-inhibition of NOTCH signaling to allow EHT. We also noted increased NOTCH1 receptor expression on the DLL4^{high} cells, while NOTCH1 expression is very low in DLL4^{low} cells. Thus, it is possible that DLL4^{high} cells can interact with NOTCH1^{low} cells. We agree that further mechanistic insights on NOTCH regulation of EHT will be essential to better understand the role of NOTCH signaling in this unique form of morphogenesis during development.

5. The model in the summary figure (Figure 6D) is intriguing, but the subdivision into a Notch1 low, Dll4- (Notch-independent) and a Notch1 High Dll4+ (Notch-dependent) population poses a problem from a Notch-signaling perspective, namely who signals to who (see also comment 4 above)? In many other situations cells with high level of ligand are juxtaposed with cells expressing high Notch receptor but low ligand levels. The authors need to carefully think about the consequences of their model with regard to trans-activation vs cis-inhibition and whether other ligands or Fringe genes are expressed. A particularly interesting case of Notch signaling involving more than one ligand type and Fringes is the tip/stalk cells in the growing endothelium.

Response: We thank reviewer for the raising a very interesting idea that should be considered. So far, we did not see differences in MFNG, LFNG, and RFNG expression in in DLL4+ and DLL4- cells by RNAseq analysis.

Our schematic diagram in Fig. 6d summarizes our findings related to outcome of hematopoietic development from DLL4+ and DLL4- cells in cultures in either the presence or absence of exogenous DLL4 ligand. DLL4 is expressed by HE underlying intra-aortic hematopoietic clusters in the AGM (Richard et al, Dev Cell, 2017) and recent mouse studies have revealed significant enrichment in pre-HSCs in the DLL4+ fraction of AGM

HE (Hadland et al. Stem Cell Reports, 2017). Thus, our in vitro findings correlate with in vivo observations and suggest that induction of HE arterialization is critical to mimic the proper specification of definitive hematopoiesis and HSC formation from hPSCs in vitro. Please see added discussion on page 17.

6. What happens if OPA-Dll4 cells are replaced with immobilized ligand (Dll or Jag) in the experiments where differentiation outcomes are assessed (Figure 2D, line 160->)?

Response: Unfortunately, our multiple attempts to assess single cell outcomes in defined conditions failed. Currently, our feeder-free and serum-free conditions can't support hematopoietic development from single cells. We did not use JAG1, because we did not observe the effect of JAG1 on blood development from HE.

Minor comments:

1. The Abstract is difficult to grasp, in particular for a more broad audience in Nature Communications. The first few sentences are somewhat abstract and in fact a bit pompous; after reading them one expects all major issues in hematopoiesis to be solved. I would favor toning in tone a bit, shorten the general part and more specifically phrase the question "in a layman's terms".

Response: We modified abstract as suggested.

2. Discussion, line 355: In current study -> In the current study....

Response: Corrected.

3. For Fig 1C, I could not find the information about the antibody used to detect activated Notch, is it a VAL-1744 ab?

Response: We used NOTCH1-ICD purified antibody from Cell Signaling Technology D3B8. All our antibodies are listed in Supplementary table S1. We also realized that we missed Western blot description in the methods section. We added this section to materials and methods.

Reviewers' Comments:

Reviewer #1:

Remarks to the Author:

Authors revised manuscript has increased in quality and clarity. Most of the issues I raised were adequately addressed by adding data, changing figures, adding supplementary figures, or adding text. A couple of minor issues remain:

1. While authors clearly described the technical issue about sorting efficiency and viability and why data only accounts for 40% of cells, this is not explained in the manuscript. Authors highlight that this is well-recognized and reported; and give a couple of references. These should be added for the benefit of scientists without advanced knowledge in cell sorting technology.

2. I fully understand the very low event number in Fig3C +DAPT condition after the explanation; however I suggest this should be clearly explained in manuscript. In fact, it might be useful to show alongside the CD4 and CD8 low event- graph, another set combining CD4 or CD8 with a more frequent marker (CD144 maybe) to avoid a flow graph without many events. These could go to supplementary data.

Reviewer #2:

Remarks to the Author:

Although aspects of this ms have been adequately revised, some issues remain that should be addressed.

1. There is still no direct evidence that arterial gene expression is required for definitive hemogenic endothelial cell specification or function. If arterial gene expression is selectively inhibited, does this change hematopoietic potential? This question has not been addressed and is not beyond the scope of this paper, given that the authors state in the Abstract..."These findings demonstrate that NOTCH-mediated arterialization of HE is an essential prerequisite for establishing definitive lympho-myeloid program...".

2. Contrary to what the authors state in their rebuttal, in the AGM in vivo, arterial genes must be downregulated to enable hematopoiesis (Lizama et al., Nature Comm 2015), and not just downregulated following EHT. This should be edited and explained in the paper, in the context of these present findings.

3. The authors' statement that a role for Notch signaling in hemogenic endothelial cell specification has never been explored is not correct. A primary role for Notch signaling has been shown in hemogenic specification in the murine yolk sac; this should be acknowledged, and the statement in the Abstract "However, the role of NOTCH signaling in hemogenic endothelium (HE) specification has not been studied."...should be changed to "However, the role of NOTCH signaling in hemogenic endothelium (HE) specification in human stem cells has not been studied.

Reviewer #3:

Remarks to the Author:

The important parts of the critique has been satisfactorily attended to.

RESPONSE TO REVIEWERS NCOMMS-17-08395A-Z.

We thank the reviewer for these critical but very constructive comments on our manuscript. In the revised manuscript, we made corresponding modifications in the text indicated by red font. Please see reply to reviewer's comments below.

Reviewer #1 (Remarks to the Author):

Authors revised manuscript has increased in quality and clarity. Most of the issues I raised were adequately addressed by adding data, changing figures, adding supplementary figures, or adding text. A couple of minor issues remain:

1. While authors clearly described the technical issue about sorting efficiency and viability and why data only accounts for 40% of cells, this is not explained in the manuscript. Authors highlight that this is well-recognized and reported; and give a couple of references. These should be added for the benefit of scientists without advanced knowledge in cell sorting technology.

As requested, we added this explanation to the text on page 8. "Due to the well-recognized fragility of hPSC-derived HE and survival after single cell sorting^{1, 28}, we found that only less than 40% of single cells formed endothelial/hematopoietic colonies. Nevertheless, the total number of colonies was consistent across each of the three NOTCH conditions, thereby indicating that the sorting experiments were not affected by differences in cell viability."

2. I fully understand the very low event number in Fig3C +DAPT condition after the explanation; however I suggest this should be clearly explained in manuscript. In fact, it might be useful to show alongside the CD4 and CD8 low event- graph, another set combining CD4 or CD8 with a more frequent marker (CD144 maybe) to avoid a flow graph without many events. These could go to supplementary data.

In order to address the reviewer's concern, we added a flow plot of total cells in Fig.3c showing CD45 vs SSC to demonstrate that the reason for the very few events in the +DAPT condition of CD4 vs. CD8 is due to the extremely low percent of CD45+ cells. This is due to the fact that the T-cell culture conditions only expand and grow T-cells, and despite plating the same number of hematopoietic progenitors into each condition, the +DAPT condition has very few blood cells. The bar graph thus shows the % of CD5+CD8+ cells gated from live cells which takes into account the decreased CD45+ cells.

Reviewer #2 (Remarks to the Author):

Although aspects of this ms have been adequately revised, some issues remain that should be addressed.

1. There is still no direct evidence that arterial gene expression is required for definitive hemogenic endothelial cell specification or function. If arterial gene expression is selectively inhibited, does this change hematopoietic potential? This question has not been addressed and is not beyond the scope of this paper, given that the authors state in the Abstract... "These findings demonstrate that NOTCH-mediated arterialization of HE is an essential prerequisite for establishing definitive lympho-myeloid program...".

To provide direct evidence that arterial gene expression is required for specification definitive hematopoiesis, we assessed whether development of lymphoid cells was affected by SOX17, which is required for acquisition and maintenance of arterial identity in embryo (Corada, M. et al. Sox17 is indispensable for acquisition and maintenance of arterial identity. Nat Commun 4, 2609 (2013)). For this purpose, we transfected hESC differentiation cultures with SOX17 siRNA at initiation of HE formation. As shown in Figure S7, inhibition of SOX17 expression markedly reduced percentages and absolute numbers of T cells produced from hESC-derived HPs, thereby providing direct evidence that arterial gene SOX17 is required for establishing definitive hematopoiesis.

We would like to emphasize that NOTCH signaling is a key pathway that orchestrates the establishment of the arterial program. Figs. 3a-3d, show that inhibition of NOTCH suppresses definitive hematopoiesis, including lymphoid potential.

Thus, we demonstrated that the most critical arterial gene SOX17, along with NOTCH signaling, are required for arterial specification of HE and establishing of definitive hematopoiesis.

2. Contrary to what the authors state in their rebuttal, in the AGM in vivo, arterial genes must be downregulated to enable hematopoiesis (Lizama et al., Nature Comm 2015), and not just downregulated following EHT. This should be edited and explained in the paper, in the context of these present findings.

Similar to findings reported by Lizama, we found that NOTCH signaling is downregulated in hematopoietic cells following EHT (see supplemental Figure S8). However, how this downregulation occurs remains an open question. It is unlikely that downregulation of NOTCH1 or DLL1 and DLL4 NOTCH ligands is required for EHT to proceed. Two studies by Irwin Bernstein's group have demonstrated that HE enriched in HSCs expresses DLL4 and that presence of DLL1 in the vascular niche is critical for HSC formation (ref 37 and 64). Based on these findings, the most likely mechanism of NOTCH downregulation is cis-inhibition of NOTCH signaling as proposed by Hadland et al (ref 64). We added discussion on page 16 to explain how downregulation of NOTCH signaling following EHT may occur:

“The exact mechanism of NOTCH downregulation at EHT stage remains unknown. Although NOTCH receptors are activated by cell surface ligands in neighboring cells (trans-activation of NOTCH), NOTCH ligands expressed by the same cell typically inactivate NOTCH signaling (cis-inhibition of NOTCH)⁶². While the response to trans-Delta is graded, cis-Delta response is abrupt and occurs at fixed threshold⁶³. Thus, it is likely that in response to trans-DLL4 signaling from OP9-DLL4, AHE upregulates DLL4 expression to the threshold level required for cis-inhibition of NOTCH signaling in its own NOTCH1-expressing AHE cells thereby allowing for EHT to proceed. This interpretation is consistent with studies in the mouse system, which have demonstrated that expression of NOTCH ligands, including DLL1 and DLL4, in the AGM vascular niche with co-expression of DLL4 and NOTCH1 on emerging hematopoietic cells is critical for HE to undergo EHT and subsequent HSC amplification through limiting NOTCH1 receptor activation by cis-inhibition^{37, 64}.”

3. The authors' statement that a role for Notch signaling in hemogenic endothelial cell specification has never been explored is not correct. A primary role for Notch signaling has been shown in hemogenic specification in the murine yolk sac; this should be acknowledged, and the statement in the Abstract “However, the role of NOTCH signaling in hemogenic endothelium (HE) specification has not been studied.”...should be changed to “However, the role of NOTCH signaling in hemogenic endothelium (HE) specification in human stem cells has not been studied.

We updated Abstract as suggested and added reference to Marcelo paper from Hirschi group (<https://www.ncbi.nlm.nih.gov/pubmed/24331925>) on page 3.

We would like to emphasize that our studies demonstrated the role of NOTCH signaling in specification of immature/primordial HE progenitors into arterial-type HE. So far, we are unaware of any in vivo studies that have reported the effect of NOTCH signaling on arterial specification of purified primordial HE. Most of the studies have addressed the effect of NOTCH signaling on cells that already acquired hematopoietic properties (i.e. capacity to form hematopoietic CFUs in semisolid medium) or on blood formation from endothelial populations composed of HE and non-HE mixture, which can obscure the direct effect of NOTCH on HE.

Reviewer #3 (Remarks to the Author):

The important parts of the critique has been satisfactorily attended to.

Reviewers' Comments:

Reviewer #2:

Remarks to the Author:

The manuscript text and figures have been adequately revised, and story should be interesting for the field.